

# An Investigation of the Impact of Canadian wildfires on US Air Quality using Satellite, Model and Ground Measurements

Zhixin Xue[1*], Nair Udaysankar[2], and Sundar A. Christopher[2]

[1]Department of Chemical and Biochemical Engineering, The University of Iowa, Iowa City, IA, USA.
[2]Department of Atmospheric and Earth Science, University of Alabama in Huntsville, Alabama, USA.
[*]Corresponding author. Email: zhixin-xue@uiowa.edu

**Abstract.**

Canadian wildfires transport large concentrations of particulate matter into the United States, leading to various impacts on surface temperature, radiation balance, visibility, and exacerbating pollution-related respiratory conditions. Using a combination of surface, satellite and numerical models, this study quantifies the increase in surface fine particulate matter (PM2.5) in the Continental United States due to long-range transported smoke from Canadian wildfires during a wildfire episode from August 9th to 25th, 2018. As a widely used indicator of surface pollution levels, satellite-retrieved AOD can provide crucial information on columnar pollution mass. However, the daily spatial coverage of satellite AOD is restricted due to cloud cover. In order to quantify the daily changes of surface pollution, we fill in the AOD gaps by utilizing simulated 10-m spatial resolution AOD from a chemistry transport model (CTM). In addition, different processes affecting smoke vertical and horizontal transport are examined using two CTM simulations. Meteorological variables associated with these processes are then selected along with the gap-filled AOD product to assess the surface pollution using geographically weighted regression (GWR) and random forest (RF) models. It is found that synoptic pressure systems dominate the horizontal transport of Canadian smoke, and the ascending air and robust winds from a low-pressure system contributes to the long distances of the transport path. Other processes affecting the vertical distribution of pollutants, including boundary layer entrainment, precipitation and terrain-induced vertical mixing are also analyzed in this paper. Our results show that Canadian fires caused a substantial increase in total PM2.5, reaching up to 13% ($62 \mu g m^{-3}$) across different US EPA regions during 2018 August wildfire event.

## 1 Introduction

Airborne fine particle mass concentrations with aerodynamic diameters less than 2.5 $\mu$m (PM2.5) exposure contributed to 4.58 million deaths globally in 2017, and ambient PM2.5 were responsible for 64.2% deaths (Bu et al., 2021). PM2.5 typically originates from combustion sources, power plants, dust, sea salt, and secondary chemical reactions, and wild-land fires are primary sources of PM2.5 in the US (O'Dell et al., 2019). For regions affected the most by wildfires, like the state of Washington (WA), an increase in daily PM2.5 of 97.1 $\mu g m^{-3}$ was found, which related to 92 more mortality cases (Liu et al., 2021). Moreover, the toxicity of PM originating from wildfire smoke is 3-4 times greater than equivalent doses of ambient PM (Wegesser et al., 2009). Health impacts vary depending upon chemical composition, which in turn varies depends upon the stages of biomass





combustion and combustion temperature (Kim et al., 2018; Aguilera et al., 2021). Previous studies show clear linkages be-
tween PM2.5 and increased mortality from diseases including Ischemic heart disease, chronic obstructive pulmonary disease,
cardiovascular disease, respiratory disease, lung cancer, chronic kidney disease, hypertension, and dementia (Chen and Hoek,
2020; Bu et al., 2021; Bowe et al., 2019). The association exists not only for high levels of PM2.5 concentrations but also for
the WHO guideline exposure level of 10 $\mu gm^{-3}$ (Chen and Hoek, 2020). Besides the health impacts, wildfires' increasing

economic impacts have become another burden on the community. From simulation results, wildfire-related economic costs
increased from $ 7 billion per year to $ 43 billion per year in 2090 (Neumann et al., 2021).

Higher fire radiative power (FRP) results in longer distances of the smoke transport due to higher injection height of the
plumes (Solomos et al., 2015), and the processes that bring these injected aerosols to the surface can be complex. Injection
heights vary for different fires, and it can reach 15km in height for certain tropical fires (Paugam et al., 2016). For a Canadian

wildfire in 2002, the injection height was estimated to be 2-6 km to match the lidar observed vertical profile from the Micro-
Pulse Lidar NETwork (MPLNET) (Colarco et al., 2004). The long-range transported elevated layers of smoke from Canadian
forest fires gradually subsided to the surface in the Eastern US through combinations of subsidence process, interception, and
subsequent entrainment process by the diurnal variations of Planetary boundary layer (PBL) (Colarco et al., 2004). Synoptic
scale pressure system controls the transport of smoke from its sources. Upper-level winds transport the smoke horizontally,

and surface high pressure is found to be related to surface pollution increase through subsidence inversions (Miller et al.,
2011). Cyclonic circulation can form a multilayer PBL which is usually associated with temperature inversion and stable
stratification, causing accumulation of pollution in the convergent zone (Jiang et al., 2021b). Interactions between mountain
terrain with synoptic high-pressure systems enhance the stability in valleys (Beaver et al., 2010), while the same terrain can
also enhance vertical mixing under unstable synoptic conditions (Lang et al., 2015). During the transport process, various

processes may also change aerosol properties. For example, thick black carbon (BC)-free coatings with BC cores are found for
long-range transport smoke particles, which increase the light absorptivity of the aged smoke (Dahlkötter et al., 2014).

Aged smoke particles can grow larger via various mechanisms, including hygroscopic growth, condensation of volatile
organic species, and coagulation process (Colarco et al., 2004; Müller et al., 2007), and the larger sizes may enhance inflam-
mogenic and cytotoxic activities of the particle (Jalava et al., 2006). Although significant increases in PM2.5 were found for

long-range transport wildfire smoke, the toxicity of organic compounds may be reduced during the transport process (Jalava
et al., 2006). Some studies found long-range transport PM positively associated with cardiovascular mortality over thousands
of kilometers from the sources (Kollanus et al., 2016; Magzamen et al., 2021). In contrast, some found no relation between
short-term elevation in PM2.5 and daily mortality (Zu et al., 2016).

In order to assess the surface pollution levels, different approaches have been used for estimating PM2.5 concentrations:

spatial interpolation (inverse distance method, ordinary kriging method), linear regression (Hoff and Christopher, 2009), multi-
linear regression (Gupta and Christopher, 2009b), geographically weighted regression (GWR) (Xue et al., 2021; Ma et al., 2014;
Bai et al., 2016; Song et al., 2014), linear mixed-effect model (Ma et al., 2016; Lee et al., 2011), chemistry transport model
(CTM) methods (Geng et al., 2015; Xue et al., 2019) and machine learning models (Hu et al., 2017; Gupta and Christopher,
2009a; Zamani Joharestani et al., 2019). Traditional approaches are outdated as newer techniques combine different methods



and add various variables with spatial-temporal information to improve their prediction accuracy Zhang et al. (2018). Due
to the growth of computing power, machine learning (or artificial intelligence) has become a major focus for estimating the
spatial-temporal dynamic distribution of surface PM2.5 concentrations Zhang et al. (2018); Sayeed et al. (2022).

Satellite retrieved AOD is an essential indicator of surface pollution level (Wang and Christopher, 2003; Geng et al., 2015).
These data are beneficial because more than a 20-year record of satellite AOD is available from polar-orbiting sensors. How-
ever, cloud cover obstruct AOD retrieval and thus affect surface PM2.5 estimations on a daily basis (Goldberg et al., 2019).
Therefore, gap-filled AOD values can provide more useful information on surface pollution variations and distribution because
most techniques that use satellite AOD to estimate surface PM2.5 require reliable data at each pixel. Fusion of AOD retrievals
from different satellite sensors largely improves the AOD coverages Ma et al. (2014). Another way is to use multiple imputa-
tion methods with inputs including cloud fraction, elevation, humidity, temperature, and spatiotemporal trends to impute the
missing AOD values (Xiao et al., 2017). Based on the linear relationships of AOD and PM2.5, AOD values estimated using
seasonal, regional-specific coefficients derived from surface PM2.5 measurements can be then interpolated (Kriging) to ob-
tain a more extensive coverage AOD product Lv et al. (2017). Comparing various gap-filling methods, the inclusion of CTM
simulations essentially improves the accuracy of gap-filling results Xiao et al. (2021).

This study is designed to assess the pollution change due to fine particulate matter in the United States transported by
smoke from Canadian wildfires and to also examine the physical processes during transport that affects surface pollution along
the path. First, by turning on and off the fire emissions in the CTM in Canada, we conduct two WRF-Chem simulations to
investigate the processes that influence the transport of remote smoke aerosols in the atmosphere. According to the analysis of
different processes, we then selected variables associated with smoke's vertical distribution, which determined the relationships
between AOD and surface pollution concentrations. Finally, by filling in the satellite AOD gaps due to cloud covers using CTM
simulations, we estimate the surface pollution increase due to the remote fires using the filled AOD along with surface PM2.5
measurements and other meteorological variables.

## 2  Data and Study Area

### 2.1  Study area

We estimated daily mean surface PM2.5 at 0.1-degree spatial resolution over Continental US (CONUS) from August 9th to
25th, 2018. The study area is focused on the US (25-50° N, 64-125° W). At the same time, we also include the Canadian
region (25-67° N, 70-140° W) when performing the WRF simulation to account for the fire emissions in Canada to investigate
the influence of pollution within the US caused by remote fire sources. Therefore, we chose August 2018 as our study period
to analyze the impact of wildfires on surface air pollution in the US based on the total fire radiative power calculation based on
our previous work (Xue et al., 2021).



## 2.2 Ground-level PM2.5 observations

We obtained the daily surface PM2.5 concentration product that uses federal reference methods (FRM, with codes 88101) from US Environmental Protection Agency (EPA) for CONUS within the study period (https://www.epa.gov/outdoor-air-quality-data/). The measured frequency of each site is different (with a measurement interval of every 1, 3 or 6 days). A total of 950 EPA sites are available with approximately 71.1% sampling for daily data within the study period. Note that we discarded all PM2.5 values lower than the established detection limit of 2 $\mu gm^{-3}$ (EPA, 2018).

## 2.3 Satellite data

AOD values retrieved from satellite observations which is a columnar value of aerosol extinction, are correlated with surface pollution under certain conditions (Wang and Christopher, 2003; Hoff and Christopher, 2009). The relation between AOD and surface PM2.5 can be expressed as the following equation (Koelemeijer et al., 2006):

$$AOD = PM_{2.5} H f(RH) \frac{3 < Q_{ext,dry} >}{4\rho r_{eff}} \tag{1}$$

From the above equation, several factors that bridge AOD and PM2.5, including: aerosol layer height H, particle effective radius $r_{eff}$, aerosol mass density $\rho$, extinction efficiency under dry conditions $Q_{ext,dry}$, and the ratio of ambient and dry extinction coefficients f(RF). Thus satellite AOD retrievals are often used as a vital indicator for estimating surface pollution (Hu et al., 2014; Xie et al., 2015).We use the 550nm AOD from the Multi-Angle Implementation of Atmospheric Correction (MAIAC MCD19A2 collection 6 product) with 1-km spatial resolution (https://lpdaac.usgs.gov/products/mcd19a2v006/). The product retrieves AOD from Terra and Aqua MODIS (Moderate Resolution Imaging Spectroradiometer), and we take the mean value of all available retrievals. Considering that thick smoke is likely misclassified as clouds, we accept all AOD with or without adjacent clouds (Xue et al., 2021; Goldberg et al., 2019). Through validation with AERONET AOD, the accuracy of the MAIAC AOD product is approximately 66% within the expected error (±0.05 ±0.1 AOD) (Lyapustin et al., 2018). The 1-km resolution AOD is gridded to 10 km by averaging all valid AOD values in 0.1-degree boxes. It is worth noting that MAIAC AOD is originally retrieved at 470nm and then computes the 550nm AOD using spectral properties (Lyapustin et al., 2018; Liu et al., 2019). It has also been reported that the uncertainties and biases increase with increasing AOD (Martins et al., 2017; Qin et al., 2021).

## 2.4 Meteorological data

The AOD-PM2.5 relationship depends on various factors, including meteorological parameters (Xue et al., 2021). Boundary layer height (BLH), 2-meter temperature (T2M), 10-meter wind speed (U10M), surface relative humidity (RH), and surface pressure (SP) were obtained from the European Centre of Medium-Range Weather Forecasts (ECMWF) Re-analysis (https://www.ecmwf.int/en/forecasts). To match the AOD data with an average value of two different times (10:30 am and 1:30 pm),



we downloaded all meteorological variables at 12 pm local time. The spatial resolution of all meteorological data is 0.25 degrees, and we use the inverse distance method to interpolate all the variables to 0.1-degree spatial resolution.

## 3 WRF-Chem model and experimental design

The Weather Research and Forecasting model coupled with Chemistry (WRF-Chem V4.2.2) is applied in this study to examine the various processes that affect the transport of smoke aerosol and estimate the pollution change in the US due to Canadian fires. This section briefly describes the WRF-Chem model, the model configuration, and model physics and then introduces

the design of the numerical experiments.

### 3.1 WRF-Chem model

WRF is a state-of-art mesoscale numerical weather prediction system that offers operational forecasting a flexible and computationally-efficient platform (Skamarock et al., 2019). WRF-Chem is an in-line atmospheric chemistry model (Powers et al., 2017) and integrates chemistry and dynamics to allow simulations of the dispersion process, aerosol-radiation interaction, aerosol-

microphysics interaction, and complex interactions between chemistry, aerosols, and physics (Powers et al., 2017).

### 3.2 Model configuration

There are several gas-phase chemistry and aerosol treatments available in WRF-Chem V4.2.2. In the current study, the Model of Ozone and Related chemical Tracers Version 4 (MOZART V4) gas-phase mechanism (Emmons et al., 2010; Knote et al., 2014) is used in combination with the MOdel for Simulating Aerosol Interactions and Chemistry (MOSAIC) 4-bin aerosol

scheme (Zaveri et al., 2008). Four size bins (0.039-0.156, 0.156-0.625, 0.625-2.500, and 2.5-10.0 $\mu m$ dry diameters) are used in the MOSAIC aerosol module for the representation of the aerosol size distribution. The PBL scheme used in our simulation is Mellor-Yamada-Janjic (MYJ) turbulent kinetic energy (TKE) scheme (Janjić, 1990, 1994) and the land-surface model scheme is the Noah Land Surface Model scheme (Chen and Dudhia, 2001). The cumulus scheme is the Grell 3D cumulus scheme (Grell and Dévényi, 2002), which performs better than other schemes (Hasan and Islam, 2018). The cloud micro-physics scheme is

the Morrison 2-momentum microphysics scheme (Morrison et al., 2005; Morrison and Pinto, 2005). Model radiation treatment utilizes the Rapid Radiative Transfer Model for General Circulation Models short-wave and long-wave radiation schemes (RRTMG,(Iacono et al., 2008)), including the aerosol radiation feedback.

Meteorological initial and lateral boundary conditions for WRF-Chem simulation are obtained from the National Centers for Environmental Prediction (NCEP) Global Data Assimilation System (GDAS) Final analysis (FNL) at 0.25-degree spatial

resolution and 3-hour temporal resolution. The initial and lateral conditions of chemical species are obtained from the Whole Atmosphere Community Climate Model (WACCM) at 0.9*1.25-degree resolution with 88 levels. All the key species associated with wildfires are included: Carbon monoxide (CO), carbon dioxide (CO2), black carbon (BC), nitrous oxide (N2O), nitrogen oxides (NOx) and so on (Mills et al., 2016). In addition, the Fire Inventory from NCAR version 2.4(FINNv2.4) is used as fire emission input for the simulation. The model simulation was conducted over Canada and CONUS region from August 9th to



25th, 2018, at 10km spatial resolution with 71 vertical layers from surface to TOA (top of atmosphere). More details of the model configurations are shown in table 1.

| Option type | Selected option |
|---|---|
| Horizontal grid resolution | 10km (550*300) |
| Number of vertical layers | 71 (39 layers below 2km) |
| Microphysics scheme | Morrison 2-moment scheme (Morrison et al., 2005; Morrison and Pinto, 2005) |
| Short and longwave radiation | RRTMG (Iacono et al., 2008) |
| Land surface | Noah-MP (Chen and Dudhia, 2001) |
| Boundary layer scheme | MYJ TKE scheme (Janjić, 1990, 1994) |
| Cumulus physics | Grell 3D (Grell and Dévényi, 2002) |
| Aerosol feedback | Yes |
| Chem-opt parameter | MOZART-MOSAIC (Emmons et al., 2010; Knote et al., 2014) |
| Meteorological data input | NCEP |
| Biogenic emissions | MEGAN |
| Anthropogenic emissions | EDGAR-2010 |

**Table 1.** Parameterizations used in WRF-Chem model simulations

## 4    Methods

This section mainly described the methods we used to estimate pollution changes in the Continental United States due to long-range transported smoke. We first described the method we used for filling the AOD gaps of the MAIAC satellite AOD product,
then described two different methods for estimating surface PM2.5. Finally, we discuss how we assess the surface pollution change due to Canadian wildfires.

### 4.1    Filling the AOD gaps

To reiterate, one of the goals of the study is to estimate daily surface PM2.5 at 0.1-degree spatial resolution. AOD alone cannot provide necessary spatial coverage due to gaps in cloud cover. Therefore, we explored two commonly used methods for filling
the AOD gaps. One commonly used method for daily AOD gap-filling problems is Kriging interpolation (Kianian et al., 2021; Singh et al., 2017). At the same time, the performance of Kriging interpolation degrades with increasing distances from the training points, which implies a limitation of the method in large areas of missing data (Kianian et al., 2021). Therefore, in the second method, we combine the Kriging method with outcomes from our CTM simulations to improve the AOD-gap interpolation.





### 4.1.1 Kriging interpolation


The ordinary Kriging (OK) method computes the estimation of an unsampled point based on the weighted average of surrounding pixels (Zandi et al., 2011). Several authors have fully described the theoretical basis of this method (Cressie, 1988; Emery, 2005), and has been proved to successfully fill in the AOD gaps for air pollution studies (Ma et al., 2014). The estimated AOD value at an unsampled location (x0) can be expressed as:

$$Z'(x_0) = \sum_{i=1}^{n} \lambda_i Z(x_i) \qquad (2)$$

where i=1,2,3...n representing for the surrounding pixels, and $\lambda_i$ is the kriging weight. A major factor that expresses the spatial dependence between neighboring points is the variogram (Arslan, 2012), which is defined as:

$$\gamma_h = \frac{1}{2n} \sum_{i=1}^{n} [Z(x_i) - Z(x_i + h)]^2 \qquad (3)$$

where $Z(x_i)$ is the AOD value at point i and $Z(x_i + h)$ is the AOD value of other points that have a discrete distance h from
point i. Previous studies have applied OK method with exponential variogram (or semi-variogram) on interpolating missing AOD values (Lv et al., 2016; Hu et al., 2019). Therefore, in this study, we use the the OK method with exponential variogram to obtain a first-stage gap-filling daily AOD over our study area.

### 4.1.2 CTM interpolation

We applied the Kriging method when sufficient AOD information is available for interpolating, whereas we used CTM in-
terpolation if there were insufficient AOD retrievals. This is because the possible undetected small-scale fire sources in the CTM may introduce more uncertainties if we feed the interpolation with CTM outputs in the small-scale "missingness" of AOD. Therefore, our gap-filling method accepts kriging interpolation in regions with a small missing portion ($<20\%$). At the same time, we feed the interpolation with CTM outputs for regions with a more significant missing portion. The details of the process are shown in figure 1. To estimate the AOD value of a location where no valid MAIAC AOD is present, We first select
a 9*9-pixel box (spatial resolution of 0.1-degree). Then we calculate the differences of $AOD_{wrf}$ between the target pixel and other points. Within the selected box region, we count the total number of pixels that satisfied two conditions: (a)have valid MAIAC AOD; (b)have a small CTM AOD difference from the target pixel ($<0.1$). When the number of satisfied points is less than 50 points, we increase the radius of the selected box and repeat the process, while if the satisfying points are more than 50, we continue to calculate the ratio of pixels with valid MAIAC AOD over the total pixels. If the ratio is larger than 80%, we
accept the estimated AOD using the OK method for the target pixel. However, if the ratio is less or equal to 80%, we computed the target AOD using a geo-weighted method considering the neighboring ratio between MAIAC AOD and $AOD_{wrf}$:

$$AOD(x_0) = AOD_{wrf}(x_0) * R \qquad (4)$$





where R is the weighted ratio which can be expressed as:

$$R = \sum_{i=1}^{n} (\alpha * w) \tag{5}$$

$$\alpha = \frac{AOD_{MAIAC}(x_i)}{AOD_{wrf}(x_i)} \tag{6}$$

$$w = \frac{1 - \frac{distance(x_i, x_0)}{bandwidth}}{\sum_{i=1}^{n}(1 - \frac{distance(x_i, x_0)}{bandwidth})} \tag{7}$$

Where $distance(x_i, x_0)$ means the distance between location $x_i$ and $x_0$, and the bandwidth is selected based on the maximum distances between point $x_i$ with valid MAIAC AOD and target pixel $x_0$ within the pre-selected box. According to above equations, we can obtain the AOD prediction for the unsampled location $x_0$.

**4.2    Estimating surface PM2.5 using the gap-filled AOD**

Having the gap-filled AOD data, we can now estimate the surface PM2.5 with more extensive spatial coverage. In this study, we tested two methods (GWR and RF) for predicting surface PM2.5 using the gap-filled AOD with other meteorological variables. Due to characteristics of the regional wildfires, we chose both methods to consider spatial variations of pollution distribution. After comparing the fitting and validation results of the two methods, we apply the method with the better performance to
estimate daily surface PM2.5.

Before we perform the prediction, different data sets need to be resampled to the exact grid resolution. Grids with 0.1-degree spatial resolution are constructed over the study region. For AOD data with 1km resolution, we average all valid AOD values that fall into the same grid. Moreover, for other meteorological variables with a 0.25-degree resolution, we apply the inverse distance method to scale up all the variables into the predefined grids. PM2.5 measurements in the same grid are averaged to
one value to obtain a 0.1-degree resolution. In order to derive the relation between PM2.5 and AOD, we select data where AOD and surface PM2.5 are both available (AOD $> 0$ AND PM2.5 $> 2.0\mu g m^{-3}$) to train the models.

**4.2.1    GWR method**

To derive the surface PM2.5 using filled AOD with other meteorological variables, we used a GWR model that we fully describe in our prior work (Xue et al., 2021). The GWR model has advantages over other methods because it estimates spatially varying
relationships. The disadvantage of GWR model, on the other hand, is that the coefficients change daily according to different spatial characteristics of surface pollution, indicating increased computational expenses. To account for varying degrees of freedom centered on different locations, Adaptive bandwidth, selected by the Akaike's information criterion (AIC), is used for the GWR model. The model can be described as:

$$PM_{2.5,i,t} = \beta_{0,i,t} + \beta_{1,i,t}AOD_{i,t} + \beta_{2,i,t}BLH_{i,t} + \beta_{3,i,t}T2M_{i,t} + \beta_{4,i,t}U10M_{i,t} + \beta_{5,i,t}RH_{i,t} + \beta_{6,i,t}SP_{i,t} + \epsilon_{i,t} \tag{8}$$



Where i represents different locations, t for different days, and $\beta$ stands for the weight coefficients for different variables. The value of $\beta$ depends on the geographical weighting of surrounding observations within the bandwidth. The weighting of each observation point decreases according to an exponential curve as the distances from the target point increase.

To test the GWR model, we must preserve only a small portion of the data, leaving most data to train the model since GWR requires an adequate number of samples, and the distribution of ground observations is uneven across the nation. Leave-one-out cross-validation (LOOCV) can be a relatively accurate way to test the model, but it requires much computational cost (Xue et al., 2021). Additionally, k-fold cross-validation with large fold numbers shows similar results (Xue et al., 2021). Therefore, we performed 100-fold cross-validation to evaluate the model performances. The entire inputs for the model are split into 100 subsets, and each time, we use one subset as testing samples while the other 99 subsets as fitting samples, after repeating this process 100 times until we test the whole data set. Finally, we evaluate the model performance by comparing the correlation coefficient ($R$), and root mean squared error (RMSE) of model fitting and cross-validation.

### 4.2.2 Random Forest method

Random forest is a non-parametric model that conducts estimations of prediction values by constructing a large number of decision trees. RF randomly divides nodes into sub-nodes for each tree, and the average estimation of different trees makes up the final results (Jiang et al., 2021a; Breiman, 1996). Of various machine learning methods, RF usually outperforms other machine learning methods due to its simplicity, diverse applications, tackling with complex cross-sensitivities among various features (Gupta et al., 2021; Jiang et al., 2021a; Zimmerman et al., 2018). The two vital parameters that affect the model performance is the tree number in the forest and feature numbers. We use 100 trees and six features in this study.

For our study period, a total of 11,942 samples are collected for training the model. The number of variables is the same as inputs for the GWR model in order to maintain consistency (AOD, BLH, T2M, U10M, RH, SP, latitude, longitude and day number). We also performed the same cross-validation as the GWR model to test the model performance so that the prediction accuracy can be compared between the two models. The model that performs better is selected to estimate the surface PM2.5 for locations where no PM2.5 observations are available based on: (a)the results of the cross-validation and (b)the differences of results (RMSE and correlation coefficients-R) between model fitting and validation. Note that a lower RMSE and higher R indicate high prediction accuracy, and a slight difference in model fitting and validation means the model is not over-fitting.

### 4.3 Assessing the pollution increase caused by Canadian fires

The model that performs better from the previous step is used for estimating the gaps in surface PM2.5. To further assess the surface pollution change due to long-range transported smoke from Canadian wildfires, we conduct a control run of WRF-Chem with all fire emissions within Canada turned off while emissions in the US are kept the same. The simulated AOD of the control run (hereafter $AOD_{control}$) is then converted to surface PM2.5 based on the relation derived from previous steps:

$$PM_{2.5,control} = AOD_{control} * \frac{PM_{2.5}}{AOD_{filled}} \tag{9}$$





where $AOD_{filled}$ is the full-coverage gap filled AOD, and PM2.5 is the estimated PM2.5 using the better performed model (GWR or RF). The differences between $PM_{2.5,control}$ and $PM_{2.5}$ are then calculated to quantify the pollution change due to long-range transported smoke.

## 5 Results

### 5.1 Smoke plumes from Canadian wildfires transport to the US

Figure 2 shows the AOD change due to Canadian wildfires from August 17th to 20th, 2018. The changes in AOD distribution with and without Canadian fires vary according to wind directions and fire sources. The AOD change during the study period varies between 0 and 2.2 based on the two WRF-Chem simulations. The change is most prominent in North Dakota and Minnesota on August 17th, and the Canadian smoke continued to move southward to Iowa on August 18th. In the meantime, a large amount of Canadian smoke increased (high AOD increase) in Northwestern US (including Washington and Montana). On August 19th, Canadian smoke over the northwestern US was transported eastward, and more smoke was brought to the central US. On August 20th, Canadian smoke moved further to southern regions due to a storm system while the AOD values decreased, due to precipitation.

Compared to the previous study, a much more significant AOD change (larger high AOD coverage areas) is found in the mid-eastern US caused by long-range transport smoke from Canada (Yang et al., 2022). The larger smoke coverage areas are associated with larger fire emissions and beneficial transport conditions. Long-range transport smoke from Canada identified based on the aerosol height information contributes 40%-60% fraction to the total column AOD in New York (Wu et al., 2018). The observed AOD (at 532nm) increase for high-layer aerosol from Canadian fires ranges from 0.18 to 0.45 in New York, which matches the simulated AOD change in Eastern US from 0.1 to 0.4 (figure 2). Our estimated AOD changes from Canadian fires are lower than the study that includes the US local fires. However, they match the observed aloft-layer AOD increase in New York (Yang et al., 2022; Wu et al., 2018), which indicates a reliable quantification of AOD analysis for Canadian fires within the US using chemistry models.

### 5.2 Associations of smoke transport with synoptic scale pressure patterns

Prior studies show that the transport of smoke aerosols is usually affected by synoptic pressure patterns. High surface pressure often indicates the accumulation of surface pollution, while successive low pressure can also increase surface pollution (Chen et al., 2008). In order to investigate the relationship between synoptic pressure patterns and the smoke transport process, we compared the horizontal and vertical distribution of Canadian smoke in the US.

According to the surface pressure map on August 17, 2018 (as shown in Figure 3a), there was a high-pressure system located in Ontario, and its influence extended into parts of North Dakota. This high-pressure system was moving eastward, and as it moved east, the descending air associated with the high-pressure system inhibited vertical convection, leading to the entrapment of smoke in the lower atmosphere. Consequently, the smoke experienced dry deposition at lower altitudes, which explains the





low AOD concentration over northeastern Minnesota in figure 2a. Smoke at higher altitudes tended to be redirected around the high-pressure system. Simultaneously, there was a low-pressure system situated in the North Wisconsin region, causing the smoke to move in a northeasterly direction towards this low-pressure area. On August 18th, as shown in Figure 3b, the

high-pressure system shifted to Quebec, and its peripheral influence extended to the Madison region. This presence of the high-pressure system resulted in the formation of a narrow corridor of low AOD distribution within the path of smoke transport, as illustrated in Figure 2b. On the same day, a low-pressure system was forming over South Dakota and Nebraska. The ascending air and the robust winds along the trough axis established more conducive circumstances for the transportation of Canadian smoke. This phenomenon is clearly illustrated in figure 2c, where the smoke expanded further to the south, strongly influenced

by the presence of the evolving low-pressure system. By August 20th, this system had transitioned eastward, enveloping the states of Iowa and Missouri, as in figure 3d. Intense rainfall and thunderstorms materialized as the low-pressure system rotated over the area. This precipitation served as a scavenging mechanism, effectively eliminating the smoke through a wet deposition process. Some portions of the smoke, however, carried by the powerful winds, could potentially impact air quality in the southern regions.

Figure 4 shows the surface PM2.5 distribution from EPA stations during the four days. The spatial patterns of surface pollution differ from the AOD change map (figure 2) in that US local fires also impact the air quality. For example, on August 17th, severe surface pollution (PM2.5 30 $\mu g m^{-3}$) occurred in the northern US, yet only North Dakota and Minnesota show noticeable AOD increase from Canadian smoke, indicating that most of the surface pollution in the northwestern US is caused by local fires. This regional pollution was exacerbated by high-pressure systems situated over Montana, which suppressed

vertical air motion, thereby causing pollution to linger in the region and consequently increasing surface pollution levels. In Figure 4b, observed on August 18th, there's a noteworthy contrast to the AOD distribution. Surface pollution, in this case, registers higher values over the high-pressure region. This disparity can be attributed to the fact that the surface pollution becomes trapped near the Earth's surface due to the calm winds associated with the high-pressure system. Thus, the presence of the high-pressure system not only impedes the movement of long-range transported smoke from Canada but also contributes

to the buildup of surface pollution. On August 19th (as shown in Figure 4c), we observe an expansion of increased surface PM2.5 levels in several states within the central United States. The high AOD concentrations over Utah and Illinois can be attributed to the influence of high-pressure systems. Conversely, the increase in PM2.5 levels in Iowa and Missouri appears to align with the transport trajectory of Canadian smoke. This observation suggests that the low-pressure system has extended the range of smoke transport further to the south, reaching Missouri, in contrast to the smoke distribution in the preceding days.

The impact of low-pressure-induced smoke transport is notably more pronounced on August 20th, as depicted in Figure 4d. In the center of the cyclone, specifically in Iowa and Missouri, there isn't a significant increase in surface pollution. This absence of a significant increase can be attributed to the wet deposition process, where the Canadian smoke, carried by the robust winds associated with the low-pressure system, experiences precipitation that effectively removes it from the atmosphere. However, in the regions along the outer boundaries of the cyclone, spanning from Colorado through Kansas and Oklahoma to northern

Texas, there is a noticeable increase in surface pollution. This is a result of the Canadian smoke carried by the powerful winds of the low-pressure system, which contributes to heightened pollution levels in these areas.



With low-pressure systems in the central US, both model outputs and surface observations show longer southward transport paths of Canadian smoke in the US. Take August 20th as an example; air flows of the extratropical cyclone in Iowa state substantially determine the transport direction of the Canadian smoke. To investigate smoke's horizontal and vertical transport, we
take the pollution distribution map of three atmospheric layers (773hPa, 850hPa, and 900hPa) to compare the moving directions of smoke (figure 5). In upper layers (773hPa), Canadian smoke intrudes into the US from two directions: northwestern US and northeastern US. These two smoke plumes meet in the central US at higher level and then split into two directions while transporting downward (as shown in the black circles at 900hPa map). The downward transport directions can be explained using a conceptual model of the extratropical cyclone (figure 5(d)). Canadian smoke follows the downward airflow: part of the
smoke moves southwesterly away from the cyclone while the rest of the pollution moves toward the cyclone and goes through wet deposition processes.

Overall, the horizontal distribution and transport of Canadian smoke are closely related to pressure systems. Surface high pressure usually facilitates the subsidence of elevated Canadian smoke pollution (with surface PM2.5 increase). In contrast, a low-pressure system tends to have the opposite effect (lifting), corresponding to longer transport distances.

### 5.3   Associations of smoke transport with PBL height

PBL height plays a vital role in determining the relationship between AOD and surface PM2.5 as it is considered the vertical limit of dilution volume of pollutants. PBL evolution regulates the dispersion and mixing of pollutants between the surface and free troposphere (Nair et al., 2012, 2011): on the one hand, during a diurnal cycle, low PBL height at night is often related to high surface pollution concentrations; On the other hand, for daily variations, PBL height is primarily governed by synoptic
pressure systems which lead to changes of surface pollution (Miao et al., 2019).

Figure 6 shows the vertical distribution of long-range transported PM2.5 dry mass (from Canada), and the black line indicates the PBL height. For the northern US, where smoke aerosols are distributed higher above ground, increasing PBL height facilitates smoke intrusion and leads to higher surface pollution. PBL height grows from early morning (18UTC) to the afternoon (21UTC), and PM2.5 concentrations also increase within the PBL. Figure 7 shows the diurnal variations of one EPA
station on different days. For days with long-range transported smoke (August 20th), the increasing PBL height in the afternoon corresponds to a higher surface pollution level. However, for days without obvious wildfire smoke transport (August 29th to September 1st), PM2.5 peaks in the early morning and late night as a result of human activity and PBL height, and it is the lowest in the afternoon due to the increase of PBL height (Manning et al., 2018).

The same pattern for southern US regions during the wildfire episode (figure 6) can be seen from the vertical distribution
map: increasing PBL height in the afternoon dilutes the PM2.5 concentrations, and thus surface pollution decreases. Note that the selected regions have no local wildfires, and all the smoke is transported from other fire sources. From the vertical distribution of smoke at different times, the elevated smoke layer stayed above 2km level for both day and night. Hence, the fire activity pattern has little influence on the smoke intrusion process. The reason for selecting different times in a day to analyze PBL's influence on surface pollution is to avoid the difference in smoke distribution on different days.





## 5.4 Associations of smoke transport with topography

The AOD-PM2.5 relationship becomes complicated in the northwestern US due to the mountainous topography (Xu et al., 2018). Figure 8 shows the vertical cross-section of the wind direction at 45 N at 18UTC over the Bighorn mountain region (Northern Wyoming) on August 16 and 19th, 2018. The plain-to-mountain Eastern wind over this region on August 16th is nearly perpendicular to the mountain range, and the upslope wind transport air upward and generates vertical updrafts above the mountain peaks, which enhances the vertical mixing process. Mountain-induced vertical mixing height can reach around four times larger than flat plain regions (Lang et al., 2015). However, the upward wind prevented air exchange between the valley and eastern plain (108.5 to 109 ° W).

However, the northwestern wind on August 18th has the opposite effect on pollution transport. The wind direction changed from northwest to southwest after passing the mountain, while the wind direction above mountain height remained the same. Therefore a lee trough is formed at the east of the mountain, which lead to a decrease in surface pressure. Thus, a surface convergence zone is formed, which corresponds to the noticeable PM2.5 accumulation on the lee side of the mountain.

## 5.5 Daily coverage of satellite AOD and simulated AOD from WRF-Chem

During the wildfire period selected in this study (17 days), we calculate the daily MAIAC AOD coverage using the number of pixels with valid AOD values divided by the total number of pixels. The AOD coverage after combining Aqua and Terra data ranges from 46% to 68% (shown in figure 10). The reason for analyzing AOD coverage is that satellite AOD is a crucial indicator for predicting surface PM2.5 levels. However, it often has missing values over fire-intense regions, which can significantly impact the accuracy of predicted surface PM values. Using Model-simulated AOD in conjunction with satellite AOD can help mitigate this issue and improve predictions.

In order to show the spatial distribution of AOD coverage, we also calculate the coverage ratio of AOD of each pixel in the 17 days (shown in figure 9). The average AOD coverage for the whole study area is around 60%. For the northeastern US, 40%-70% of the days are covered by clouds, while more than 80% of the days have valid AOD values in the western US.

The model simulated 550nm AOD has a very similar distribution as the satellite AOD. Comparing MAIAC AOD with corresponding simulated AOD pixels, the correlation coefficients range from 0.3 to 0.63, and the RMSE is within the range of 0.2-0.4 (shown in table S2). Time series of the coverage and statistics are shown in figure 10. There are no clear correlation between MAIAC AOD coverage and the correlation of two AOD products, as the missing AOD areas for each day are randomly distributed affected by satellite swath coverage and cloud contaminations. Overall, simulated AOD is lower than satellite AOD potentially due to underestimation of fire emissions, especially for small-scale fires (Wiedinmyer et al., 2011). FINNv2.4 identified fire sources based on the combinations of thermal anomalies product of MODIS and VIIRS (Visible Infrared Imaging Radiometer Suite) (Wiedinmyer et al., 2011; Li et al., 2021). Adding high spatial resolution fire detection information from VIIRS increases the total burned area by 280% compared to the previous FINN version that used MODIS detection only (Li et al., 2021). However, thick smoke and cloud cover primarily affect the detection of fires (Fu et al., 2020; Schroeder et al., 2014), causing AOD underestimation in regions with missing fire detection.



The comparison between WRF simulated AOD and satellite retrieved AOD shows high spatial correlation, indicating similar smoke pathway between the model and satellite observations. We also compare the ground AERONET AOD with the

model simulated values. The correlation coefficient during the 17 days between two AOD product is 0.54, and the RMSE is around 0.06. Similar to the comparison with satellite AOD, model simulated AOD values show underestamations compared to AERONET AOD.

### 5.6 AOD gap filling

Our gap-filled AOD increased the mean coverage from 60% to 92%, and missing values of the gap-filled AOD were mainly

distributed at the edges of our study area due to limited satellite retrieval to derive accurate $AOD_{filled} - AOD_{MAIAC}$ relationships.

To illustrate the importance and differences of feeding the interpolation with CTM outputs, we choose August 13th to display the differences when fires are detected near cloud edges. Figure 11 shows the AOD distribution of the CTM-filled AOD, kriging-filled AOD, and MAIAC AOD. The south-central US region (including Texas, Oklahoma, and Nebraska) has

large missing AOD areas due to clouds. The kriging-filled AOD for this region is evenly distributed with values around 0.3. However, the CTM-filled AOD shows more variations and a clear smoke transport path along the wind direction. The primary reason for this difference is some small-scale fires detected near the cloud edges in Oklahoma. According to the fire emission document (https://www.acom.ucar.edu/Data/fire/data/finn2/README_FINNv2.5_Feb2022.pdf), both 375m resolution VIIRS fire detection and 1km resolution MODIS thermal anomalies are used for estimating fire emissions. This enhancement in

fire detection provides a more accurate estimation of surface pollution in the presence of clouds. Compared with the surface PM2.5 distribution of this same day, we find the same distribution pattern as our CTM-filled AOD: high PM2.5 ($> 20 \mu gm^{-3}$) distributed in central Texas all the way to Eastern Oklahoma. Therefore, CTM-filled AOD provides closer patterns as observed surface pollution distribution at regions with large cloud covers. Our results indicate the inadequacy of kriging methods in such cases.

### 5.7 Daily PM2.5 estimation

Figure 12 shows the GWR model fitting and cross-validation results. The colors represent the probability function that determines the possibility that the ground-level measurements and the estimations from the GWR model are equal to each other. Lighter color means a higher concentration of samples. The R and RMSE of the GWR fitting model are 0.85 and 6.2 $\mu gm^{-3}$, respectively, and the R and RMSE of the 100-fold cross-validation are 0.8 and 7.2 $\mu gm^{-3}$. The difference between model

fitting and cross-validation is relatively small, which means that the model is not overfitting and the prediction accuracy is stable. The slope of the cross-validation best-fit line (solid black line) is 0.72, indicating that the model slightly underestimates the surface PM2.5, and the biases increase with increasing AOD. The reason for this underestimation is sample biases: 1) a limited number of stations in the Western US; 2) EPA station distribution in the Western US mainly concentrates in the two populated regions, showing extremely uneven distribution. Only 3% surface PM2.5 observations exceed the unhealthy limit





(PM2.5 $35.5\mu gm^{-3}$) during the 17 days, and among these stations, 62% of the observations came from stations located in Washington and California.

Figure 13 shows the results for RF fitting and cross-validation. RF model fits well with the training samples with R of 0.99 and RMSE of 1.86, while the cross-validation results degrade significantly (R=0.85, RMSE=6.2). The slope of the validation best-fit line is 0.7, which is of similar values to the GWR method. The difference between model fitting and cross-validation

indicates that the model is over-fitting and has limited prediction accuracy for this case. One possible reason for this could be the limited number of training samples with average daily available measurements of around 700 during the study period (Jiao et al., 2021). A space-time RF model can be utilized to enhance RF model performance (Wei et al., 2019). Compared with GWR, the RF model showed slightly higher prediction accuracy although it is less stable.

Though the difference in cross-validation results for RF and GWR is slight, the daily pollution variation estimated from the

two methods shows completely different trends. Figure 14 shows the mean PM2.5 variation over the three highly polluted areas (EPA Region 8,9,10) during the 17-day period calculated from GWR, RF, and EPA ground stations. For region 8, EPA stations measured two pollution peaks during the study period: one peak on August 19th and another smaller peak on August 24th. Mean PM2.5 from the RF method also captured the two peaks on the same day, while the regional mean values were slightly lower than the measurements. However, GWR has a different peak on August 11th but no noticeable increase for the remaining

days. For EPA Region 9, both RF and EPA stations show a decreasing trend for the first few days and then slowly increase with time, while GWR has two clear peaks, which are not shown for the other two methods. For the most polluted region 10, the highest peaks from August 19th to 22nd are shown in EPA and RF method, but GWR shows low values for the same period. The differences between GWR and RF methods come from the distribution of sample points. Most samples in EPA region ten are distributed in Washington state, which indicates that pollution increase in WA can easily lead to an increase for the

whole region 10 when calculating the mean PM2.5 for the region. At the same time, the GWR method sometimes captures the variations for other regions though only limited measurements are available. Evenly distributed samples may increase the estimation accuracy a lot for both methods. According to the variation trends of these polluted areas, we choose RF to estimate the surface PM2.5 in this study.

Our RF daily mean estimation of surface PM2.5 during the 17-day wildfire event for each EPA region ranges from 3.3 to 62

$\mu gm^{-3}$ (shown in table 2). The mean surface pollution is highest at region 10 while lowest at region 1, which is comparable with previous study (Xue et al., 2021).



| Date | Region1 | Region2 | Region3 | Region4 | Region5 | Region6 | Region7 | Region8 | Region9 | Region10 |
|---|---|---|---|---|---|---|---|---|---|---|
| Aug-9 | 7.6 | 8.9 | 8.2 | 7.1 | 12.2 | 9.3 | 11.6 | 20.9 | 28.5 | 31.0 |
| Aug-10 | 7.0 | 9.6 | 9.7 | 7.6 | 14.0 | 10.5 | 15.6 | 18.8 | 24 | 32.5 |
| Aug-11 | 6.4 | 8.0 | 9.4 | 8.8 | 16.8 | 11.2 | 17.4 | 18.2 | 20.5 | 25.9 |
| Aug-12 | 5.8 | 6.9 | 9.1 | 10.4 | 15.8 | 13.7 | 17.0 | 18.3 | 18.5 | 25.1 |
| Aug-13 | 5.5 | 6.3 | 8.1 | 12.1 | 15.4 | 14.4 | 14.4 | 18.3 | 15.2 | 30.5 |
| Aug-14 | 5.0 | 6.5 | 8.7 | 12.7 | 14.2 | 14.5 | 13.9 | 17.3 | 15.6 | 31.9 |
| Aug-15 | 7.9 | 14.3 | 13.4 | 13.1 | 13.0 | 13.6 | 12.3 | 16.2 | 14.3 | 34.8 |
| Aug-16 | 9.8 | 15.7 | 14.2 | 11.0 | 11.3 | 11.0 | 11.1 | 17.8 | 13.3 | 33.2 |
| Aug-17 | 10.8 | 14.1 | 12.9 | 8.2 | 12.7 | 10.1 | 11.4 | 17.9 | 15.3 | 31.0 |
| Aug-18 | 6.4 | 8.8 | 7.8 | 7.0 | 14.6 | 9.5 | 13.7 | 17.6 | 16.2 | 33.5 |
| Aug-19 | 6.3 | 7.8 | 8.3 | 7.6 | 12 | 9.5 | 16.1 | 24.5 | 16.3 | 47.3 |
| Aug-20 | 5.9 | 7.7 | 7.6 | 7.1 | 13.2 | 11.9 | 14.5 | 15.8 | 16.8 | 61.7 |
| Aug-21 | 5.8 | 6.4 | 6.9 | 7.6 | 12.0 | 12.6 | 16.3 | 16.8 | 22.1 | 46.3 |
| Aug-22 | 3.3 | 7.2 | 8.5 | 9.8 | 10.8 | 12.0 | 15.1 | 15.1 | 18.6 | 43.5 |
| Aug-23 | 5.8 | 8.1 | 9.0 | 11.5 | 12.1 | 9.2 | 13.9 | 18.8 | 17.1 | 45.4 |
| Aug-24 | 11.8 | 11.1 | 9.0 | 11.3 | 9.5 | 8.8 | 14.8 | 21.4 | 16.4 | 35.5 |
| Aug-25 | 11.4 | 9.7 | 10.0 | 11.7 | 11.3 | 9.1 | 14.1 | 17.9 | 16.7 | 23.1 |

**Table 2.** Mean PM2.5 concentrations ($\mu g m^{-3}$) over different EPA regions (1-10) estimated using RF method

## 5.8 Pollution change due to long-range transported smoke from Canada

By applying the coefficients derived from estimated PM2.5 and WRF-Chem AOD on AOD values simulated from the WRF-Chem control run, we estimated the PM2.5 increase from Canadian wildfires. Table 3 shows the daily mean PM2.5 increase for different EPA regions. The regional mean PM2.5 increases range from 0.001 to 7.7 $\mu g m^{-3}$, and the contribution of Canadian wildfire smoke to total PM2.5 ranges from 0.02% to 13%. Region 10 is most affected by Canadian fires, while region 6 is the least. The contribution of Canadian smoke to total PM2.5 in Northeastern US (regions 1,2,3) can be even larger than in northwestern US (regions 8 and 10) on some days. During the wildfire episode, the contribution of wildfire smoke to total PM2.5 closely followed the wildfire activity pattern. However, due to the transport distances, it usually took about 24 hours for the smoke to move from region 10 to region 1. Therefore, the pattern of change in PM2.5 in the Eastern CONUS lagged that of Western CONUS by a day.

Due to the CAA, the overall trend of surface PM2.5 in the recent years is gradually decreasing (figure S1). The decreasing trend is evident in August mean PM2.5 calculated from EPA surface observations from EPA region 1 to region 7, but variations of PM2.5 in region 8 to 10 is more affected by the occurrence of wildfires. For years with low wildfire occurrences (FRP <





10000 MW), August surface PM2.5 show a steady pattern in region 8 and region 10 without any apparent rising or dropping. PM2.5 in region 9 shows descending pattern in August in years without large fires, while it can reach up to three times more of the baseline in wildfire years. For regions 1 to 7, August mean PM2.5 concentrations decrease about 5% -10% each year in different regions. Compared with table S1, long-range transported smoke from Canada compensates for the descending trend in a negative fashion. For region 8 to 10, wildfires (combining local and remote fires) increases August mean surface PM2.5

by 7% - 16%.

In conclusion, due to the concurrent local and remote wildfires, the long-range transport smoke contributed to about half of the surface pollution increase in EPA regions 8, 9, and 10. For other EPA regions, Canadian smoke compensated the CAA, causing surface pollution to rise.

| Date | Region1 | Region2 | Region3 | Region4 | Region5 | Region6 | Region7 | Region8 | Region9 | Region10 |
|------|---------|---------|---------|---------|---------|---------|---------|---------|---------|----------|
| Aug-9 | 6.8 | 6.7 | 9.7 | 8.3 | 8.9 | 10.0 | 9.3 | 7.0 | 5.6 | 9.4 |
| Aug-10 | 6.1 | 10.7 | 11.2 | 9.1 | 4.3 | 3.8 | 8.5 | 3.3 | 7.8 | 3.1 |
| Aug-11 | 1.7 | 3.5 | 3.6 | 6.5 | 4.7 | 3.9 | 5.4 | 7.0 | 8.1 | 4.0 |
| Aug-12 | 8.6 | 6.8 | 4.0 | 3.8 | 4.0 | 4.4 | 4.5 | 2.7 | 5.5 | 2.1 |
| Aug-13 | 0.2 | 0.02 | 0.3 | 1.9 | 4.4 | 2.7 | 3.3 | 6.0 | 5.0 | 5.4 |
| Aug-14 | 6.0 | 5.9 | 3.9 | 3.0 | 4.5 | 3.0 | 4.4 | 3.3 | 1.8 | 4.1 |
| Aug-15 | 2.6 | 3.4 | 5.5 | 3.5 | 3.1 | 1.0 | 2.4 | 5.6 | 4.5 | 5.5 |
| Aug-16 | 3.3 | 4.0 | 3.4 | 2.1 | 6.8 | 2.8 | 4.3 | 5.1 | 4.0 | 5.5 |
| Aug-17 | 7.7 | 8.0 | 5.4 | 3.9 | 5.0 | 3.4 | 3.4 | 3.1 | 5.3 | 7.7 |
| Aug-18 | 3.1 | 3.9 | 3.3 | 2.4 | 7.0 | 1.7 | 5.2 | 8.9 | 6.1 | 6.4 |
| Aug-19 | 9.8 | 8.4 | 7.7 | 4.5 | 10.8 | 4.4 | 6.3 | 5.3 | 2.3 | 3.7 |
| Aug-20 | 8.6 | 7.0 | 3.5 | 4.0 | 4.7 | 3.3 | 1.8 | 9.2 | 6.1 | 12.4 |
| Aug-21 | 0.3 | 5.5 | 4.8 | 2.1 | 11 | 4.3 | 8.1 | 12 | 4.9 | 11 |
| Aug-22 | 12.2 | 11 | 9.5 | 8 | 10 | 6 | 8.8 | 8.6 | 6.5 | 11.6 |
| Aug-23 | 11.6 | 13.1 | 6.3 | 9.1 | 5.9 | 2.5 | 6.2 | 7.5 | 10.8 | 9.3 |
| Aug-24 | 6.1 | 5.1 | 7.5 | 6.1 | 6.5 | 10.6 | 9.1 | 9 | 6.4 | 7.5 |
| Aug-25 | 12.6 | 9.3 | 5.5 | 5.2 | 6.5 | 7.3 | 5.2 | 8.7 | 4.8 | 8.5 |

**Table 3.** Percentage increase of PM2.5 concentrations (%) due to Canadian wildfires over different EPA regions (1-10) estimated using RF method

## 5.9 Uncertainties and limitations

The main uncertainties and limitations of this study come from the CTM model and various inputs of the model, and the surface pollution estimation model also leads to some uncertainties:



1) Since satellite fire detection is affected by various factors, including cloud cover, fire sizes and the background environment, emission inputs for the WRF-Chem simulations derived from the fire detection products introduces uncertainties into our simulations and create regional biases in the simulated AOD values.

2) In addition to the fire detection biases, assumption made in computing fire emissions also cause the following uncertainties that affect the simulations: a)fire sizes and duration; b)amount and distribution of biomass fuels; c)fraction of different emissions from the biomass fuel Soares et al. (2015). These factors may influence the mass concentration and distribution of smoke aerosols.

    3) The unevenly distributed EPA stations primarily affect the performance of the two pm2.5 estimation models, causing
completely different daily variation trends of regional mean PM2.5. Therefore, the model performance may be improved if we use more EPA stations (We use FRM monitors only in this study).

    4) Since we need the relationships between satellite AOD and CTM AOD to calculate the filled AOD values, the filling values cannot be derived if the area of missing satellite AOD is larger than the radius thresholds we set for deriving the relationships. For days with large areas of missing satellite AOD at the boundary region of our study region, we sometimes have missing
AOD values at the boundary. This can be improved by increasing the radius thresholds or the study region to leave space for the boundaries.

## 6   Conclusion

This study first analyzed the influence of different physical processes on the transport of long-range transported smoke aerosols by comparing two WRF-Chem simulations with and without Canadian wildfires. Then we utilized the simulated AOD from
CTM and Kriging interpolation with a geographically weighted method to fill in the daily AOD retrieval gaps caused by cloud covers. Then we estimated the surface PM2.5 concentration using GWR and RF methods and tested the two predictions using cross-validation and trend analysis to choose a better-performing method. Finally, by turning off the Canadian wildfire emissions in the CTM simulations, we calculated the surface PM2.5 concentrations from the CTM AOD outputs using the coefficients derived from previous estimations. The differences in PM2.5 of the two estimations indicated the change brought
by long-range transported smoke from Canadian wildfires. The main findings of our study are:

    1) Synoptic scale pressure systems are the dominant drivers of horizontal pollution transport pathways. In the meantime, the pressure systems can also affect the vertical distribution by ascending or descending smoke. Under most circumstances, the subsidence flow of high-pressure systems facilitates the drifting process of elevated smoke layers and thus increases surface pollution. In contrast, the cyclonic storm system leads to a longer transport path (further south of CONUS in this study) and
directs the elevated smoke to the ground in different directions. Therefore, the co-occurrence of low-pressure systems and smoke aerosols corresponds to larger pollution areas.

    Certain weather patterns can facilitate the dispersion of smoke, hence, during severe wildfire events, it becomes crucial to closely monitor the meteorological conditions and alert the public about the potential risks of deteriorating air quality, even if the fires are hundreds of miles away. Furthermore, when conducting prescribed fire burning, it is essential to take into account



the possibility of smoke being transported to densely populated areas. In other words, proper consideration of the weather conditions and potential wind patterns is crucial to minimizing the impact on nearby populations.

    2) PBL height negatively correlates with surface pollution in most cases with pollution sources below the boundary layer. In contrast, it positively correlated with surface pollution when the bottom of the long-range transported smoke layer drifted close to the boundary layer.

3)Through analyzing the vertical motion over the Bighorn mountain region, we found that the flow interactions with complex terrain play and important role in modulating the vertical distribution of particulate.When pollution enters the lower atmosphere, Westerly winds may cause pollution accumulation in the eastern plain, while easterly winds can enhance vertical mixing, decreasing pollution over the eastern plain.

    4) Daily AOD coverages combining Aqua and Terra MODIS range from 46% to 68% during our study period, and our filled
AOD values using CTM AOD outputs are able to fill in the missing gaps.

    5) Daily PM2.5 estimations using the filled AOD product with other meteorological data using the RF method (R=0.85) perform better than the GWR model (R=0.8), and the RF model captures the daily variations of different EPA regions calculated from EPA stations.

    6) Regional mean ratio of total surface PM2.5 concentrations that came from Canadian wildfire smoke ranges up to 13%,
and EPA Region 10 is most affected by Canadian fires while region 6 is the least. The PM2.5 change pattern in Eastern CONUS often lags the Western CONUS by a day.

    Our study found that presence of synoptic scale pressure systems leads to a higher proportion of CONUS region being affected by long-range transported smoke from Canada. Typical airflow patterns that are associated with extratropical cyclones are particularly effective at transporting elevated layers of smoke to the surface and fanning the associated particulate pollution
over large areas. Such transport pathways associated with extratropical cyclones need to be considered when forecasting for the effects of smoke pollution from Canadian wildfires on vulnerable populations in the CONUS region. Our study also further confirms findings from prior research (Xue et al., 2021) that show wild fires becoming the dominant source of particulate pollution as the effects of industrial pollution continue to recede due to stringent regulations.



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



**Figures**

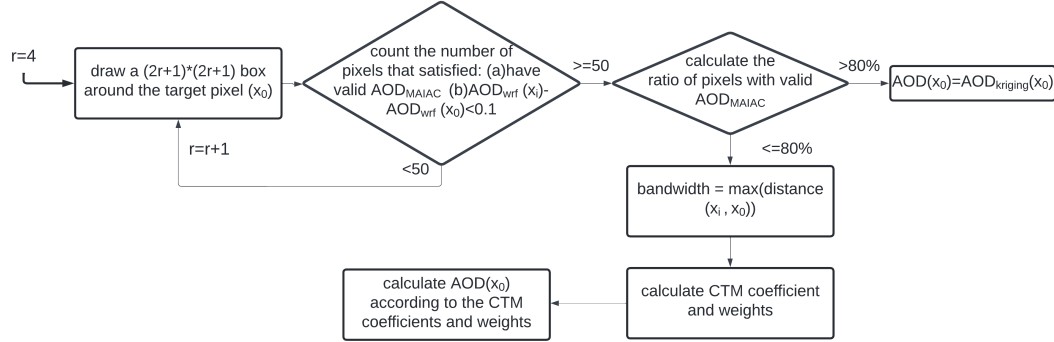

**Fig. 1.** *flowchart of the AOD gap-filling process*

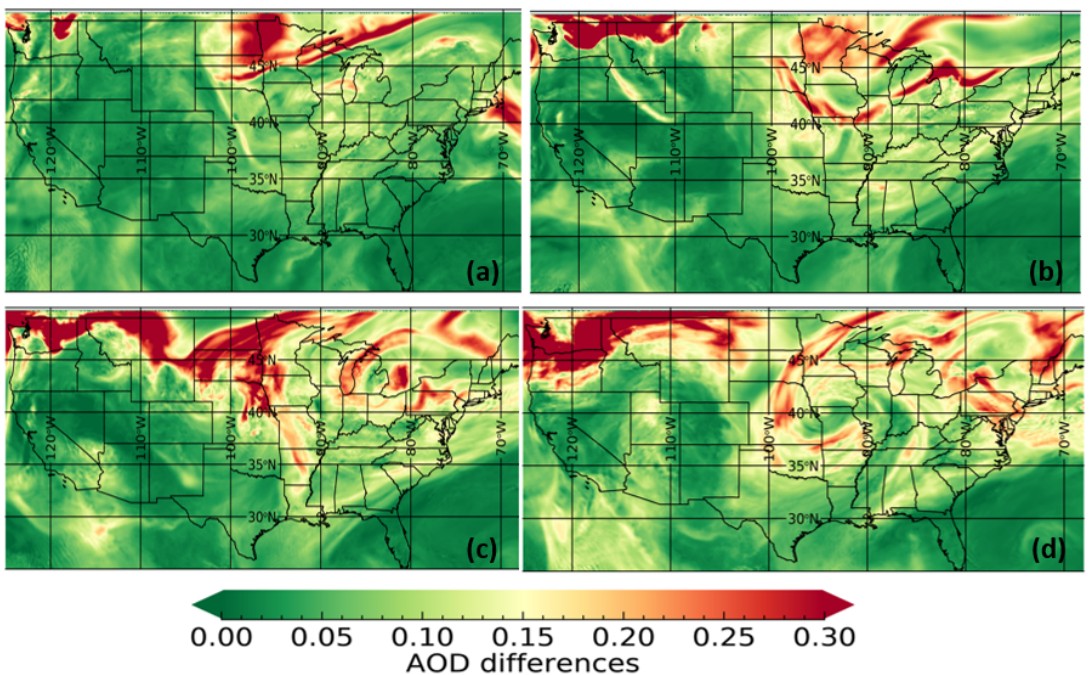

**Fig. 2.** *AOD change due to Canadian wildfires from August 17th to 20th, 2018: (a)August 17th,2018; (b)August 18th,2018; (c)August 19th,2018; (d) August 20th,2018. The AOD change is calculated using the difference of two WRF-Chem simulations.*



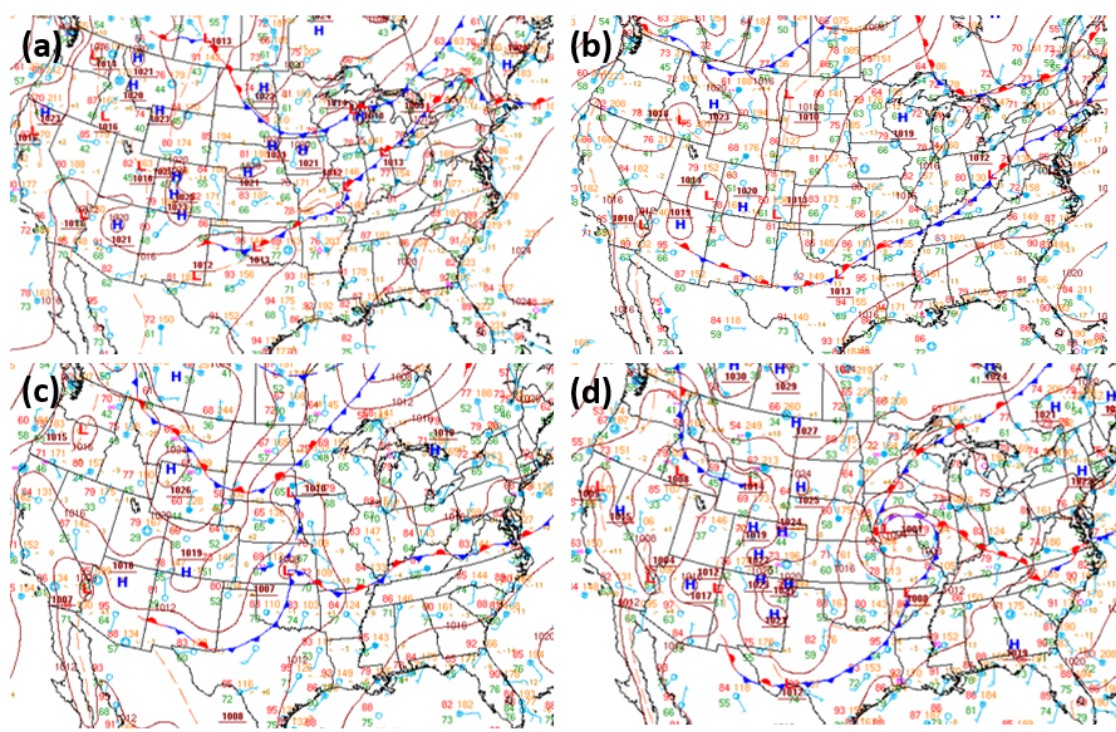

**Fig. 3.** *Surface weather maps from August 17th to 20th, 2018 (https://www.wpc.ncep.noaa.gov/archives/web_pages/sfc/sfc_archive.php): (a)August 17th,2018; (b)August 18th,2018; (c)August 19th,2018; (d) August 20th,2018.*



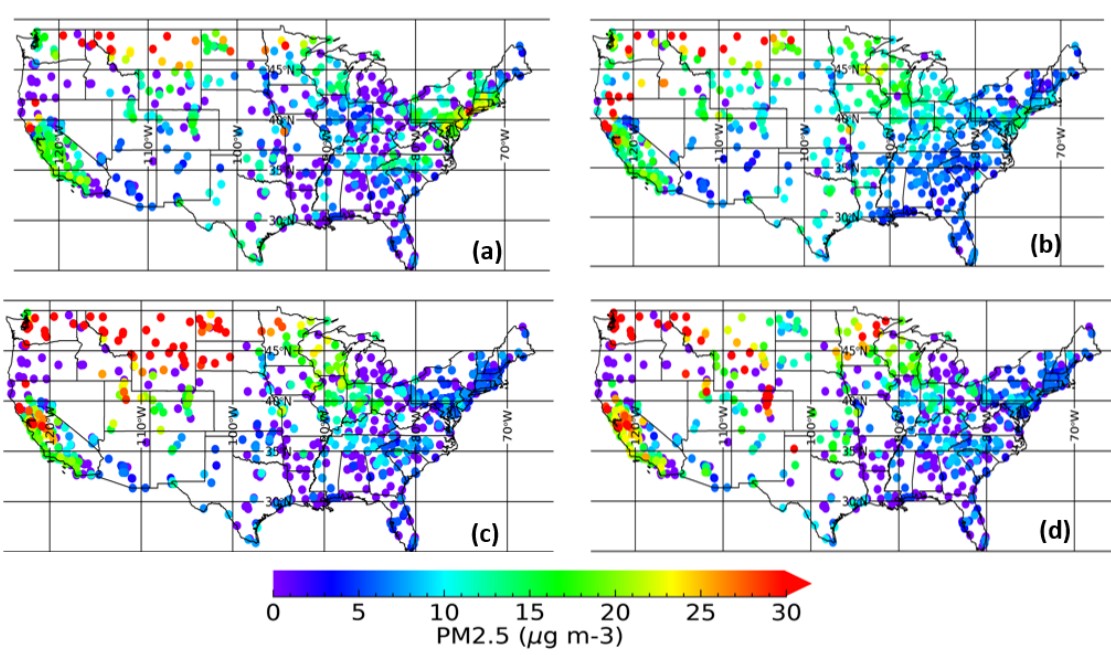

**Fig. 4.** *Surface PM2.5 measurements from EPA stations over CONUS from August 17th to 20th, 2018: (a)August 17th,2018; (b)August 18th,2018; (c)August 19th,2018; (d) August 20th,2018.*



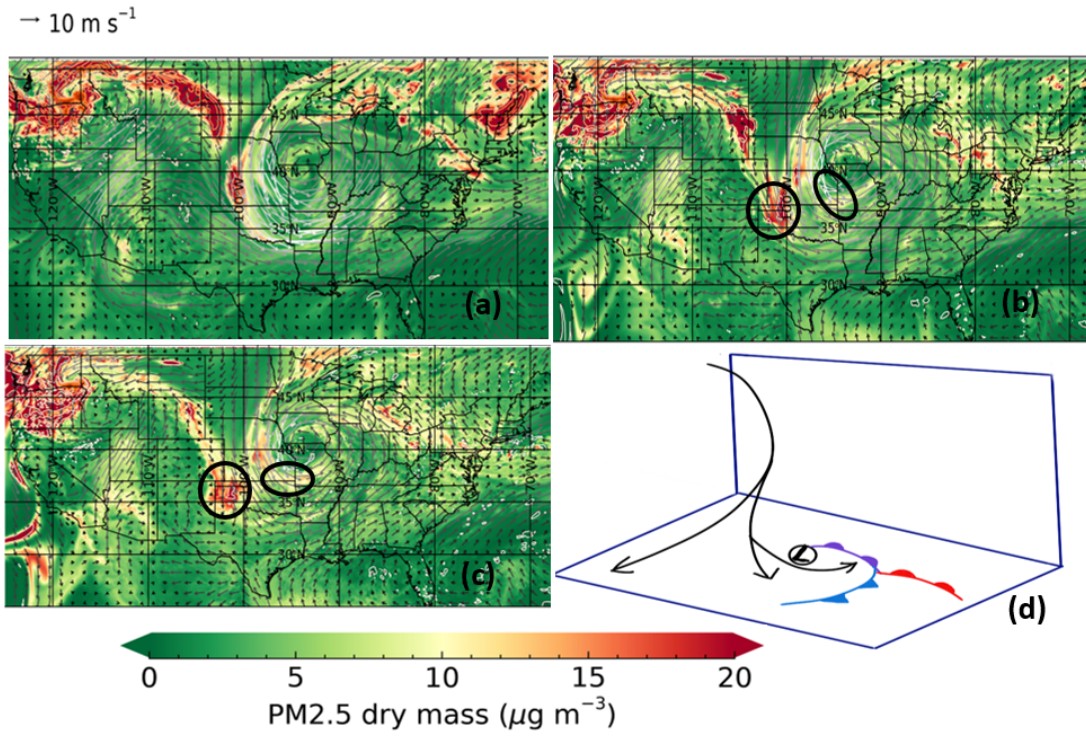

**Fig. 5.** *PM2.5 dry mass distribution on August 20th,2018 at different levels (a)773hPa (b)850hPa (c)900hPa (d)conceptual model of airflow patterns in a typical extratropical cyclone based on (Cotton et al., 2011).*





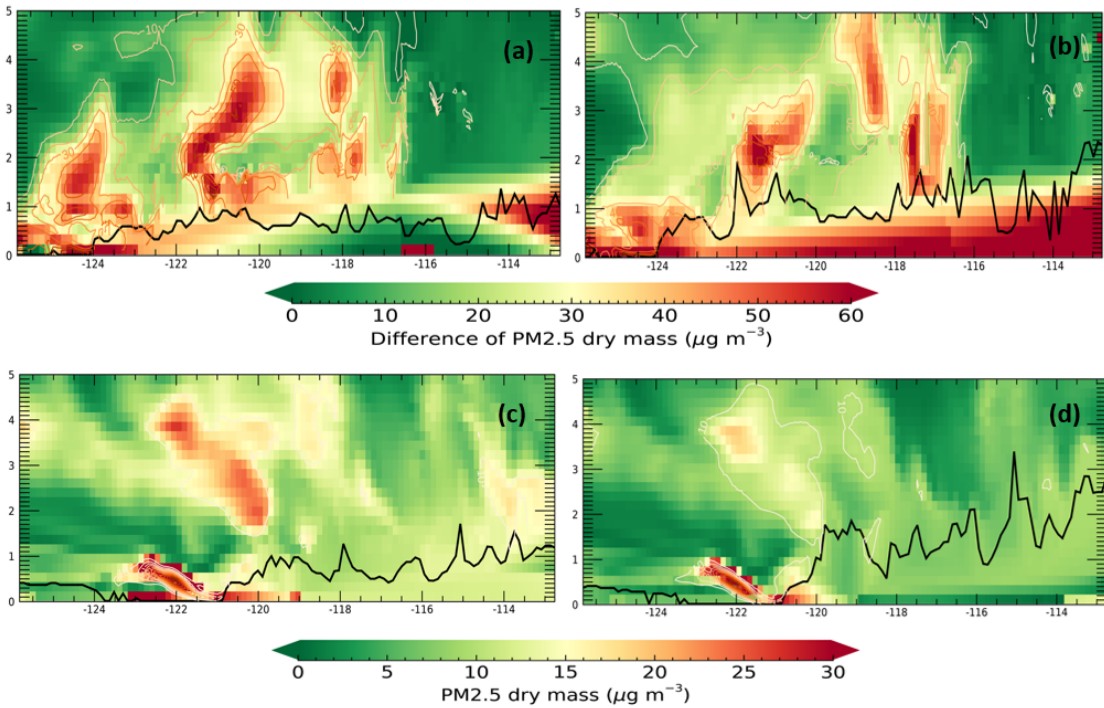

**Fig. 6.** *Simulated vertical cross section of PM2.5 dry mass with PBL height in 45 ° N (first row) and in 35 ° N (second row) at 18UTC (first column) and at 21UTC (second column).*

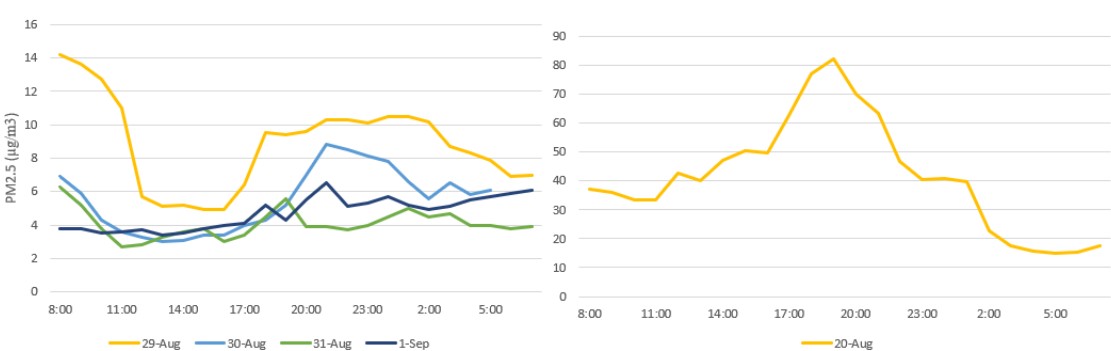

**Fig. 7.** *Diurnal variations of surface PM2.5 concentrations from one EPA station (39.7 ° N, 104.9 ° W) on different days.*




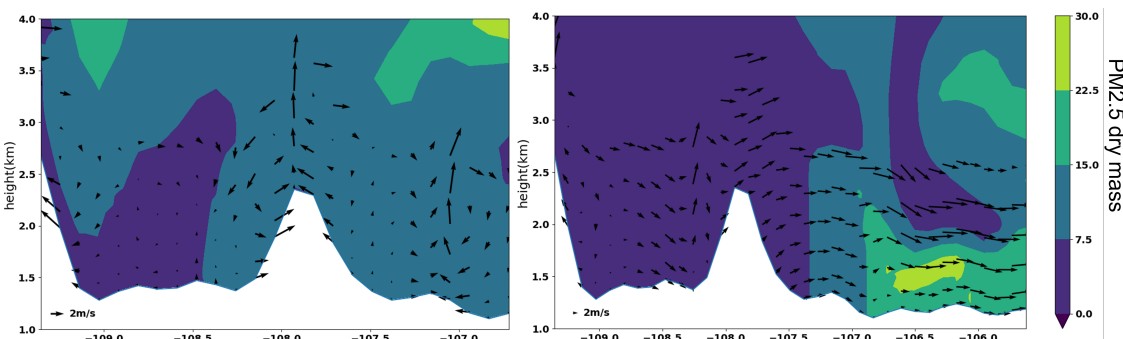

**Fig. 8.** *Simulated vertical cross section of PM2.5 dry mass with wind direction and terrain height in 45 ° N on August 16th (left) and 19th (right) at 18UTC.*

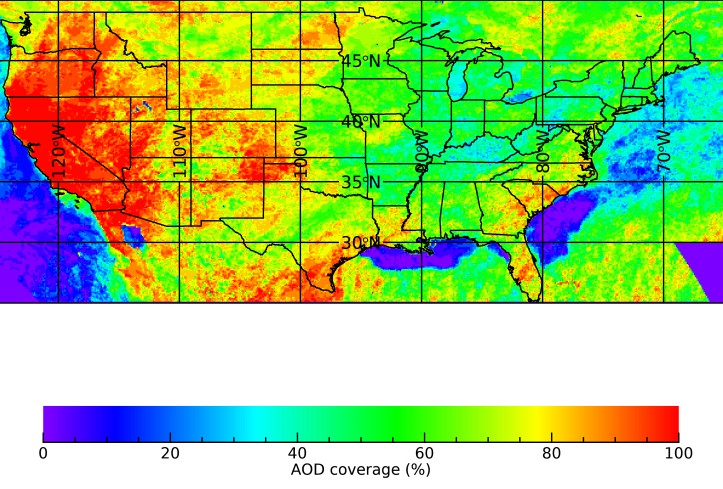

**Fig. 9.** *AOD coverage during the 17-day period*



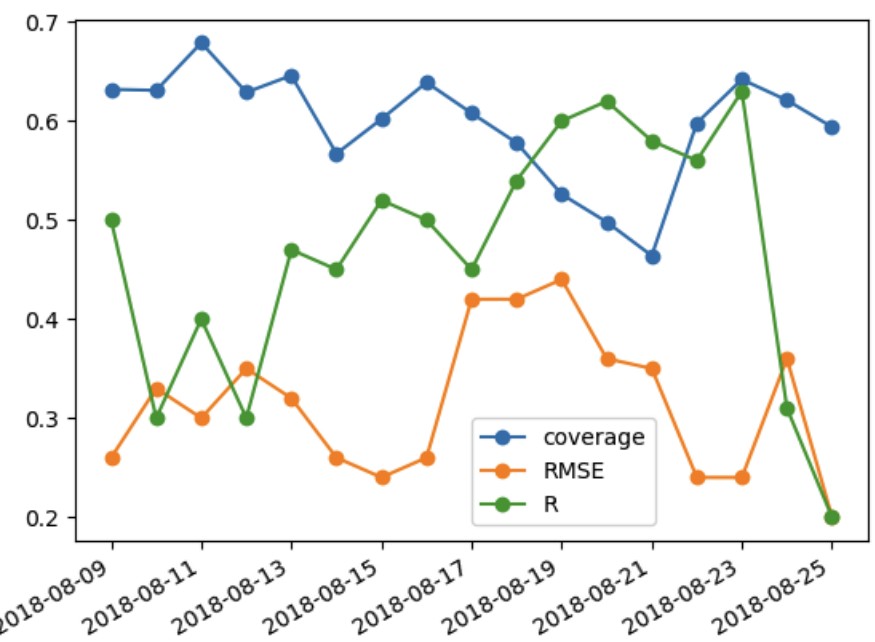

**Fig. 10.** *Time seires of MAIAC AOD coverage (blue line), correlation coefficient (green line) and RMSE (orange line) of two AOD products (WRF-CHEM AOD and MAIAC AOD).*

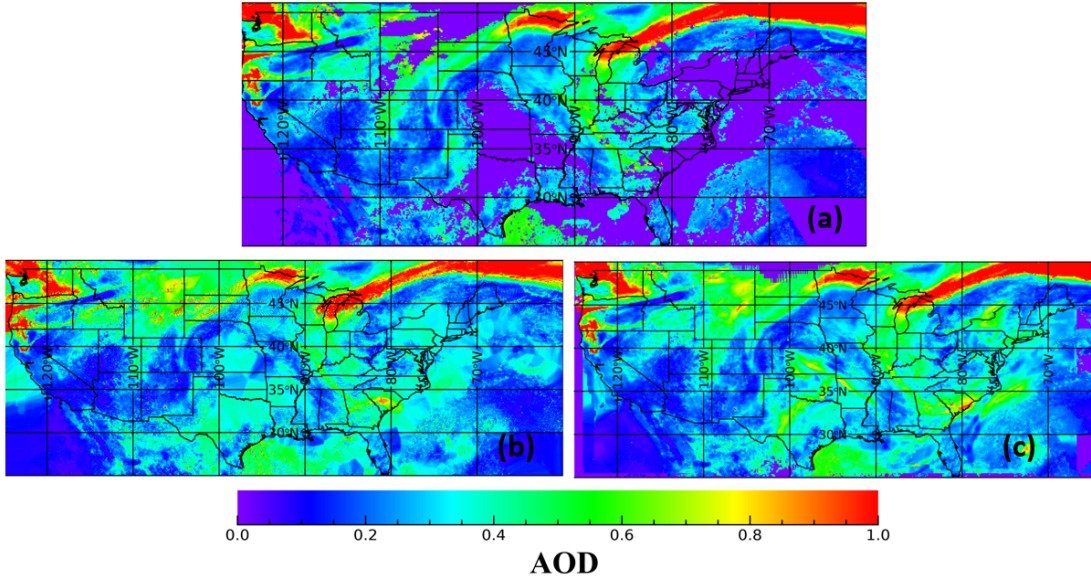

**Fig. 11.** *AOD distribution from (a) MAIAC, (b) Kriging method and (c)CTM-filled AOD on August 13th 2018.*



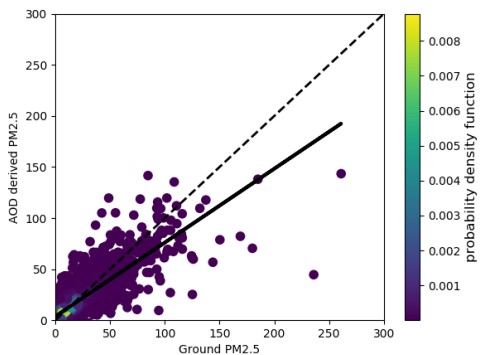

**Fig. 12.** *Fitting and k-fold cross validation results for GWR method*

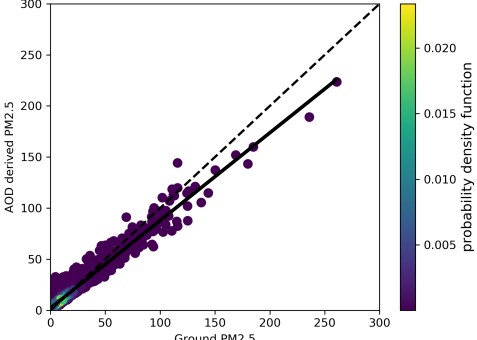
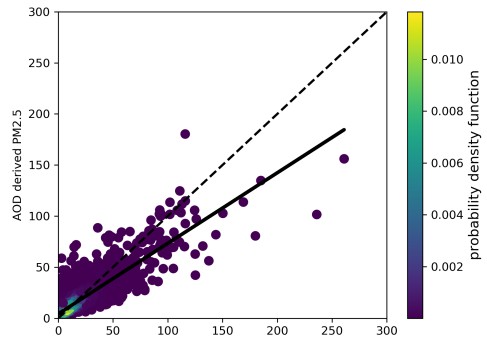

**Fig. 13.** *Fitting and k-fold cross validation results for RF method*



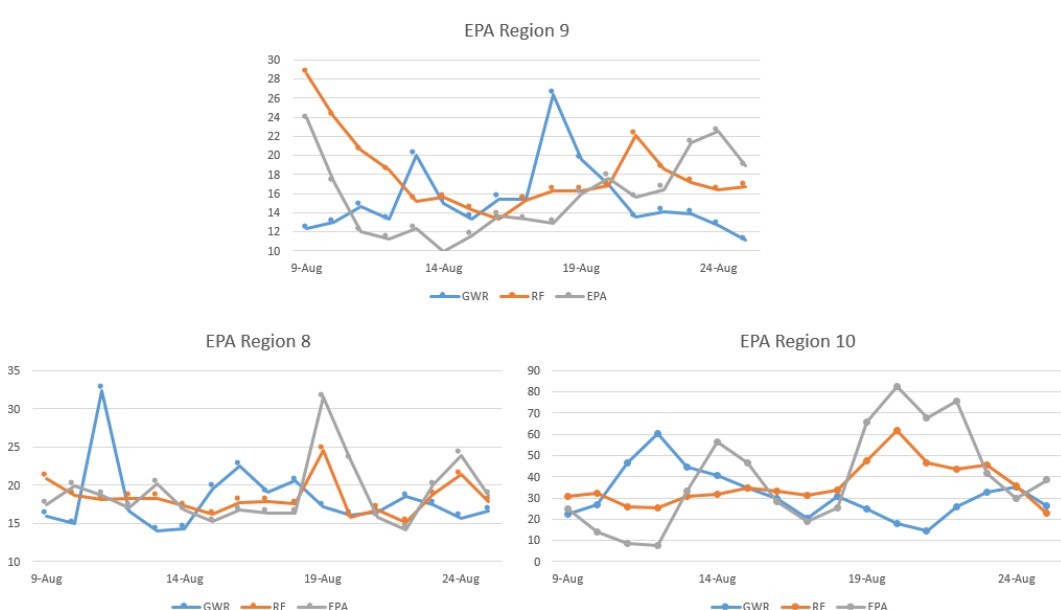

**Fig. 14.** *Mean PM2.5 concentration variations over the top 3 polluted areas (EPA region 8,9,10).*





*Author contributions.* ZX and SC conceived and designed the study. ZX performed the data analysis and prepared manuscript. ZX, SC and NU contributed to editing the manuscript and provided valuable suggestions on data analyss and interpretation.

*Competing interests.* The authors declare no competing interests.

*Acknowledgements.* We acknowledge use of the WRF-Chem preprocessor tool (mozbc, fire_emiss, bio_emiss and anthro_emiss) provided by the Atmospheric Chemistry Observations and Modeling Lab (ACOM) of NCAR.