# Peer review of "An Investigation of the Impact of Canadian wildfires on US Air Quality using Satellite, Model and Ground Measurements: supplemental document"

_EGUsphere, 2024_

## Author Response (AR1)

Reviewer 1

This manuscript demonstrates the application of a method that combines information from surface monitors, satellites, and a 3D model to estimate wildfire smoke impacts over a 17-day time period. There are significant issues with the writing and some aspects of the science, so I feel the manuscript requires major revisions before it can be published.

Major comments
In general, the writing is imprecise and there are quite a few typos. I've pointed out many examples in the specific comments. These issues caused the review to take about twice as long as it would have if the authors had edited their manuscript better. It should not be the reviewers' responsibility to edit writing mistakes and imprecision, and it adds to the stress on the peer review system.

We sincerely apologize for the writing issues and typos in the manuscript, which placed an additional burden on your review. We have carefully revised the manuscript to improve precision and clarity, addressing all specific comments and conducting a thorough proofreading to ensure quality. Thank you for your time and effort in reviewing our work.

Section 4.3 and equation 9. The method will fail when a significant fraction of the smoke impact is out of the PBL. For example, if the AOD without smoke is 0.1 and the AOD with smoke is 1.0, but 100% of the smoke is in the free troposphere, the real impact of removing smoke should have no effect on surface PM2.5. However, equation 9 would attribute 90% of the surface PM2.5 to smoke (as opposed to the correct value, 0%). This erroneous calculation could have a large impact on the results in Table 3, which has some of the main results of the paper that show up in the abstract.

Thank you for raising this point. We agree that the $\frac{PM_{2.5}}{AOD}$ ratio may not accurately represent the smoke contribution to surface PM2.5 when a significant fraction of smoke resides above the PBL. To address this, we have revised our method to avoid directly using the $\frac{PM_{2.5}}{AOD}$ ratio. Instead, we now estimate the surface PM2.5 for the control case using the ratio between the WRF-Chem-simulated surface-level P2.5 concentrations of the experiment run (with Canadian smoke) and control run (without Canadian smoke), as follows:

$$PM_{2.5,control} = PM_{2.5} * \left( \frac{PM_{2.5,WRF,control}}{PM_{2.5,WRF,experiment}} \right)$$

The revised method uses the modeled surface P2.5 concentrations from WRF-Chem for both the control and experiment cases, which inherently accounts for the vertical distribution of smoke. By doing so, it avoids the assumption of directly using AOD to estimate surface-level P2.5. We have updated the Section and equation 9, the results in the abstract have also been updated accordingly.

Sections 5.1-5.4 seem very scattered, and I am not currently seeing how they contribute to the main points of the paper. I recommend removing some of these sections and better integrating the remaining parts with the following sections.

Thank you for your valuable feedback. To enhance the coherence of the manuscript, we have removed Sections 5.3 and 5.4 as suggested and integrated the remaining content more effectively with the subsequent sections.

How is the modeled AOD defined in a cloudy column? Aerosol scattering blows up as f(RH) approaches 100%. How is this handled in a cloudy gridbox and how do the assumptions affect the results?

AOD calculation in WRF-Chem is based on chemical species and does not directly account for the effects of RH or cloud droplets. Aerosol optical properties calculations in WRF-Chem rely on a sectional approach and assume dry aerosol conditions (Barnard et al., 2010). As mentioned in section 3.2, we used four size bins (0.039-0.156, 0.156-0.625, 0.625-2.500, and 2.5-10.0 $\mu m$ dry diameters) in the simulation. The particle mass and particle number are simulated separately for each bin. Importantly, since the bins are defined based on dry particle diameters, water uptake or loss does not cause particles to transfer between bins. In each bin, the particles are assumed to be spherical and internally mixed. Given this simplification, the conversion from chemical to optical properties follows the following steps:

1. Aerosols are divided into size bins, and the chemical masses $M_{i,j}$ and particle number $N_i$ are calculated for each size bin, $i$ is the bin number and j is the chemical species.

2. The volumes of these chemicals are computed for each bin $V_{i,j} = \frac{M_{i,j}}{\rho_i}$ $(unit: \frac{cm^3}{cm^3\ dry\ air})$

3. The physical diameter of each bin can be calculated using: $D_{p,i} = 2\left(\left(\frac{\sum_j V_{i,j}}{N_i}\right)/\left(\frac{4}{3}\pi\right)\right)^{\frac{1}{3}}$, the aerosol size distribution is defined by $N_i$ amd $D_{p,i}$

4. Now that we have the size distribution, the bulk refractive index of the particles in a bin can be computed, $m_{s,i}$ and $m_{c,i}$ are the bulk complex refractive indices of the shell and core for bin "i". The core refractive index is assigned the value of 1.85+0.17i, and the shell refractive index is given by:

$$m_{s,i} = \frac{\sum_{j=1,j\neq BC}^{j} m_j V_{i,j}}{\sum_{j=1,j\neq BC}^{j} V_{i,j}}$$

5. Shell/core Mie theory is then used to find the absorption $Q_{a,i}$ and scattering efficiency $Q_{s,i}$, then other optical properties such as absorption $B_{abs}$ and scattering coefficient $B_{scat}$ can be calculated to get the AOD values:

$$B_{scat} = \sum_{i=1}^{8 \; bins} N_i Q_{s,i} \pi \left(\frac{D_{p,i}}{2}\right)^2$$

$$B_{abs} = \sum_{i=1}^{8 \; bins} N_i Q_{a,i} \pi \left(\frac{D_{p,i}}{2}\right)^2$$

$$B_{ext} = B_{scat} + B_{abs}$$

$$AOD = \int B_{ext}(z)dz$$

Details of the "aerosol chemical to AOD" calculation processes can be found in (Barnard et al., 2010).

There are several assumptions can affect the results:
1. The chemical mass inputs for AOD calculation depend on the emission inventory used (e.g., FINN in our study). Errors or biases in the fire emissions inventory can directly propagate into the AOD estimates.
2. The AOD in cloudy grid boxes is effectively a dry AOD because RH effects are not included in the calculations. Post-processing is needed to account for hygroscopic growth, using model output for aerosol water content and the hygroscopicity parameter to estimate how scattering increases with RH. Neglecting hygroscopic growth can lead to an underestimation of AOD, particularly in humid or cloudy regions.
3. The model assumes spherical, internally mixed particles, which may not accurately represent real-world aerosol shapes or external mixing, potentially causing errors in optical property calculations.
4. Due to assumptions about particle size distributions, overestimation of optical properties can occur for coarse-mode particles, particularly in bins with diameters >2.5 μm.

Specific comments
Title: Is "measurements" also applying to "satellite"? If not, then it should be "satellites". If yes, then it's confusing to have "model" in the middle.

We have revised the title for clarity. The new title is: "An Investigation of the Impact of Canadian wildfires on US Air Quality using Model, Satellite and Ground Measurements".

L8: 10-m grid spacing is impressive.

Thank you for catching this error. We have corrected it to 10-km grid spacing.

L16: unclear what these numbers mean. 62 ug m-3 would be far more than a 13% increase considering typical non-smoke concentrations in the US are <10 ug m-3.

To clarify, the number here represents the PM2.5 increase caused by Canadian wildfires only during the study period. There were also local wildfires in the US contributing to the total PM2.5 concentrations, but this value isolates the impact of Canadian wildfire smoke on PM2.5.

Based on the major comments above, we recalculated the surface PM2.5 for the control case. As a result, the PM2.5 increase values have been updated in the revised manuscript to ensure accuracy and clarity.

L22: 92 mortalities over what time period?

The time period is the summer of 2020. We have revised the sentence for clarify as follows: "For regions affected the most by wildfires, like Washington state, an increase in daily PM2.5 of 97.1 $\mu g m^{-3}$ during the summer of 2020 was found, which related to 92 more mortality cases."

L23: I'd say "has been found to be 3-4 times greater" given that this is just one study and smoke and non-smoke composition varies.

Thank you for the suggestion. The sentence has been revised accordingly to reflect this point.

L25-28: Is this sentence about wildfire smoke PM2.5 or PM2.5 in general?

This sentence refers to PM2.5 in general. We have reorganized the paragraph to ensure a more logical flow.

L30-31: Using past tense in a sentence about the future. I'd say "have been projected to increase from…".
Thank you for catching this in consistency. The sentence has been corrected to: "From simulation results, wildfire-related economic costs have been projected to increase from $7 billion per year to $43 billion per year in 2090."

L32 and in general: Focus on making the topic sentence be an intro to the full paragraph. This paragraph moves a long way from FRP and injection heights.

We have revised the first sentence to better introduce all the contents of the paragraph and adjusted the paragraph for clarity and coherence. The revised paragraph now reads:
"
The transport and dispersion of wildfire smoke are influenced by multiple factors, including fire intensity, injection height, atmospheric dynamics, and terrain interactions. Higher fire radiative power (FRP) results in longer smoke transport distances due to higher plume injection heights. A global analysis of over 23,000 wildfires found that significant injection heights into the free troposphere are primarily observed in the boreal forests of North America and Siberia during the northern summer. Smoke remnants from Canadian stand-replacing forest fires have been observed at altitudes exceeding 13 km. Once injected, smoke plumes can descend to the

surface through a combination of subsidence, interception, and diurnal entrainment within the Planetary Boundary Layer (PBL), as observed in the Eastern US for 2002 July Canadian forest fire event (Colarco et al. 2004).

Atmospheric circulation plays a key role in shaping smoke transport. Upper-level winds facilitate long-range horizontal transport, while surface high-pressure systems enhance ground-level pollution through subsidence inversions (Miller et al. 2011). Cyclonic circulation can form a multilayer PBL, characterized by temperature inversions and stable stratification, which trap pollutants in convergent zones (Y. Jiang et al. 2021). Interactions with mountain terrain further modulate smoke dispersion: under stable synoptic conditions, valleys become more stagnant, while unstable conditions promote vertical mixing (Beaver et al. 2010; Lang, Gohm, and Wagner 2015)."

L33-36: Boreal injections have been found to be higher on average than tropics in the MISR plume height climatology (https://www.mdpi.com/2072-4292/10/10/1609), so the contrasting of limited cases seems to give the wrong impression.

Thank you for pointing this out. To avoid confusion, we have included the global analysis of injection heights from (Val Martin et al., 2018) to provide a broader context. Additionally, we used a case study of a Canadian wildfire event to illustrate that boreal wildfire in North America, particularly during the northern summer, generally exhibit higher injection heights. This clarification helps to highlight the regional significance of boreal forest fires without overstating individual cases.

L47: Volatile organic compounds, by definition of being volatile, do not condense. Some of the oxidation products of VOCs can condense. Also, the citations here are not dedicated investigations of the specific processes listed in this sentence, though these studies do exist.

Thank you for your comments. We agree that the original terminology could be misleading. What we intended to convey is that as smoke cools away from the flame front, some semi-volatile gases, including oxidation products of volatile organic compounds (VOCs), can condense onto existing emitted particles, forming organic or inorganic coatings (Junghenn Noyes et al., 2021). These coatings increase particle size and can significantly affect aerosol optical properties. We have revised the sentence for clarity as follows:

During transport, smoke particles undergo physical and chemical transformations that influence their sizes, such as hygroscopic growth (Carrico et al., 2005; Gomez et al., 2018), SOA formation (Ahern et al., 2019) condensation of semi-volatile species, and coagulation process (Aloyan et al., 1997; Sun et al., 2019).

L54-59: Of what data sources?

Spatial interpolation methods rely on ground-based PM2.5 measurements, while linear regression models use satellite-derived aerosol optical depth (AOD) along with surface PM2.5 measurements. Multi-linear regression methods typically combine surface PM2.5 measurements, satellite AOD, and meteorological datasets from models. Geographically weighted regression (GWR) and machine learning approaches draw from similar data sources as multi-linear regression, integrating surface PM2.5, satellite AOD, and meteorological inputs to capture spatial and temporal variations in PM2.5 concentrations.

L59: Which of the approaches above are the traditional ones?

The traditional approaches mentioned here refer to spatial interpolation and linear regression methods, which have been widely used in earlier studies for estimating PM2.5 concentrations. We revised this paragraph to clarify:

To assess surface pollution levels, various approaches have been used for estimating PM2.5 concentrations, each utilizing different data sources. Traditional methods such as spatial interpolation (e.g., inverse distance weighting, ordinary kriging) rely primarily on ground-based PM2.5 measurements, while linear regression methods combine satellite-derived AOD with surface PM2.5 measurements (Hoff and Christopher 2009). More advanced methods, like multi-linear regression, typically incorporate surface PM2.5 data, satellite AOD, and meteorological datasets from models (Gupta and Christopher 2009b). Geographically weighted regression (GWR) and machine learning methods use similar data sources as multi-linear regression but offer better spatial resolution and adaptability to local variations (Xue, Gupta, and Christopher 2021; Ma et al. 2014; Bai et al. 2016; Song et al. 2014). Linear mixed-effect models add temporal variability by including both fixed and random effects (Ma et al. 2016; Lee et al. 2011), while chemistry transport models (CTMs) leverage detailed atmospheric chemistry and physics simulations using emissions inventories, meteorological fields, and chemical species distributions (Geng et al. 2015; Xue et al. 2019). Traditional approaches, such as spatial interpolation and linear regression, are limited in their ability to combine different mechanisms and add various variables with spatial-temporal information to improve their prediction accuracy, which newer techniques address (Zhang, Rui, and Fan 2018). Due to the growth of computing power, machine learning (or artificial intelligence) has become a major focus for estimating the spatial-temporal dynamic distribution of surface PM2.5 concentrations (Zhang, Rui, and Fan 2018; Sayeed et al 2022).

L63: Strange to say that AOD is an *essential* indicator of surface pollution since it is a column measurement. There are much more direct ways to get surface PM (e.g., an in situ measurement), so AOD is not essential. The PM2.5:AOD ratio should be discussed in this paragraph.

We agree that AOD is not a direct indicator of surface PM2.5 and have revised the text to clarify that AOD serves as a valuable proxy, particularly in regions or periods with limited in situ measurements. We have also expanded the discussion to include the PM2.5:AOD relationship, emphasizing how it helps link columnar AOD measurements to surface PM2.5 concentrations.

L85-87: This sentence makes it seem like there are 2 different WRF-Chem domains, implying 2 different simulations. In the later WRF-Chem description, it seems like this is not the case.

Thank you for pointing out this potential confusion. We have revised the sentence to clarify that the study area corresponds to the inner domain, while the larger outer domain includes remote fire sources, both within a single WRF-Chem simulation.

"The study area (inner domain) focuses on the US (25–50°N, 64–125°W), while the outer domain of the same WRF simulation extends to Canada (25–67°N, 70–140°W) to account for Canadian fire emissions and their contribution to US pollution from remote fire sources."

L116: T2M, U10M, and SP are not directly related to the AOD:PM2.5 ratio. I would guess that including them could lead to overfitting.

The selection of predictor variables is based on our previous study (Xue et al., 2021), where we successfully estimated surface PM2.5 using the same set of variables (AOD, BLH, T2M, U10M, RH, SP) for the same time period (August 9$^{th}$ to 25$^{th}$, 2018). In that study, we performed leave-one-out cross-validation (LOOCV) to assess overfitting. The difference in $R^2$ between the fitting and validation was minimal (0.037), suggesting that overfitting was not a concern. Therefore, we assumed the same approach would be valid for this study.

L128: Should "in-line" be "online"?
L130: Both aerosols and clouds have microphysical processes (condensation, coagulation, etc.), so please say "aerosol-cloud interaction" here. Similarly, since aerosols have chemical and physical processes, what does "interactions between chemistry, aerosols, and physics" mean here?

The original sentence, "WRF-Chem is an in-line atmospheric chemistry model" Is correct. As referenced in (Powers et al., 2017): "WRF-Chem is a WRF-based in-line atmospheric chemistry model."
To clarify and avoid any potential misunderstanding, we have revised the sentence to: "WRF-Chem is an atmospheric chemistry model that fully integrated with the meteorological framework of WRF, enabling the simulation of various chemical and physical processes related to aerosol transport, including dispersion, aerosol-cloud interactions, and other key mechanisms."

L140: 2-moment

Corrected.

L146: "All the key species… are included" is a bold statement. Are you sure?

To avoid potential misunderstanding, we have revised the sentence to: "Key species associated with wildfires are included…"

Table 1 or accompanying texts: What are the lat-lon limits of the domain?

We have added a description to clarify the lat-lon limits of the two nested domains in the accompanying text:
"The model simulation was conducted over two nested domains: an outer domain covering the Canadian region (25-67°N, 70-140°W) and an inner domain focused on the CONUS region (25-50°N, 64-125° W). "

L176: Double "the"

Corrected.

Section 4.1.2: How is AOD defined in the CTM in a cloudy column (see major comment)?

Please refer to our reply to the major comments above for a detailed explanation.

L180-182: This sentence is confusing.

We revised the sentence to:
" We applied the Kriging method for interpolation in areas with sufficient AOD information, while using CTM interpolation where valid AOD retrievals were unavailable, to minimize uncertainties associated with small-scale "missingness" of AOD. These uncertainties arise because CTM relies on fire inventories derived from satellite fire detection products, which may fail to capture small-scale fires due to the spatial resolution of satellite observations and fractional fire coverage within a pixel (Fu et al., 2020). Such undetected fire sources can lead to inaccuracies in CTM outputs. To better represent the AOD distribution in regions with small-scale "missingness," we prioritize the Kriging method over CTM interpolation."

Section 4.2: Please explain how variability in plume height is captured in the 2 methods in this section.

In our study, plume height is indirectly accounted for through the inclusion of input variables. GWR method captures spatial variability by weighting observations based on their geographic proximity. While GWR does not explicitly account for plume height variability, it reflects its effects spatially through the relationship between AOD and PM2.5.
RF method, on the other hand, captures the non-linear relationships between input variables and surface PM2.5. It incorporates the influence of plume height variability on PM2.5

concentrations, as AOD is a columnar measurement linked to the vertical distribution of aerosols.

In both methods, Plume height is not directly modeled as a standalone variable, but its effects are incorporated through the chosen predicters.

Section 4.3: See major comment.

Please refer to our reply to the major comments above for a detailed explanation.

L264-265: Is this other study investigating the exact same time period? If not, is the comparison relevant?

The other study referenced here investigates a wildfire event from June 6–12, 2015, while our study focuses on August 2018. Although the time periods differ, the comparison remains relevant as both studies explore the impact of Canadian wildfire smoke on the US. The comparison highlights the consistent role of long-range transported Canadian smoke as a significant pollution source during wildfire seasons. By examining AOD values in different years, we provide a broader context for understanding the recurring influence of Canadian wildfires on US air quality.

L268-272: These several sentences were all confusing.

We have revised these sentences to clarify that the observed AOD increases in the US caused by Canadian smoke are recurring phenomena during wildfire seasons. While the magnitude of AOD increases varies across events and years, these examples serve to provide important background context and underscore the widespread and significant impact of Canadian wildfire smoke on air quality in the US. The revised text now better reflects this focus.

Figure 3. It's very hard to read the text on this figure. Can we see well enough what we should be taking from the maps? Would something like 500 hPa geopotential heights be better for showing the steering of smoke? Should the maps include Canada?

Thank you for your feedback. To address this, we have changed the surface weather maps in Figure 3 for easier visualization and improved clarity. We hope the updated figure better conveys the key information.

L281: Is there an estimate of dry dep? Normally dry dep is fairly slow for accumulation-mode sizes.

Estimation of dry deposition or sensitivity tests for its parameterization is beyond the scope of this study. In our simulations, aerosol dry deposition follows the scheme by Binkowski and Shankar (Binkowski & Shankar, 1995), where dry deposition velocity is calculated using gravitational settling velocity ($V_g$), aerodynamic resistance ($R_a$), and surface resistance ($R_s$).

This parameterization has been found to overestimate deposition for accumulation-mode aerosols (Emerson et al., 2020; Ryu & Min, 2022). While several improved schemes address overestimation of deposition velocity, these improvements are more effective for larger aerosols (e.g., 2.5–10 μm in the fourth size bin) and have limited impact for smaller aerosols (diameter <2.5 μm) (Ryu & Min, 2022).

Section 5.3 and 5.4. These just show special cases for specific locations while the rest of the paper shows smoke affects more broadly. I don't think these sections add much, and the manuscript may benefit from cutting them.

Thank you for your feedback. To enhance the coherence of the manuscript, we have removed Sections 5.3 and 5.4 as suggested and integrated the remaining content more effectively with the subsequent sections.

Figure 6 needs axis labels. Also, neither the caption nor the text says which day it's for. Figure 10 gets discussed before Figure 9.

We have corrected the order of the figures in the manuscript so that Figure 9 now appears before Figure 10.

L365: "AOD is crucial" same comment about the use of "essential" earlier.

We have revised the sentence to:
Analyzing daily AOD coverage is essential, as satellite-derived AOD serves as a columnar indicator of pollution with extensive spatial coverage and high-resolution data, making it a valuable predictor for estimating surface PM2.5 concentrations alongside other variables.

L372-374: It's not clear to me that these are actually good-enough performance values, particularly the RMSEs. Can you give more context to these values?

The comparison between model AOD and satellite AOD in our study focuses on a high-pollution period (wildfire events), a broad spatial domain (entire CONUS region), and fine temporal resolution (daily). These factors inherently introduce additional challenges in achieving high agreement, as model AOD often struggles to accurately capture satellite-observed AOD due to various uncertainties in model simulations, including fire emissions, aerosol properties, and transport processes. This is why we do not directly rely on model-simulated AOD for surface PM2.5 estimations in our analysis.

For context, a study of 550 nm AOD comparing MODIS satellite data with WRF-Chem AOD for eastern North America during August 2008, using a 12 km spatial resolution, reported an R value of 0.689 (Crippa et al., 2017). In contrast, our R values range from 0.3 to 0.63. While this may seem lower, it is important to note that our comparison is performed under more challenging conditions, including daily temporal resolution and the presence of extreme events such as wildfire smoke. These factors naturally lead to increased variability, and thus slightly lower R values are expected.

Additionally, in a study evaluating wildfire smoke during the Williams Flats fire (August 4–8, 2019), R values for comparisons between modeled and MAIAC AOD ranged from 0.1 to 0.5, with RMSE values between 0.1 and 0.3 (Ye et al., 2021). However, the Williams Flats fire was relatively small compared to the Canadian wildfires in August 2018 examined in our study. Furthermore, their study domain was limited to a smaller 10° × 5° region, while ours spans the entire CONUS, making our study inherently more complex.

These comparisons emphasize the additional challenges faced in our analysis. While our R values and RMSE reflect the difficulties of modeling AOD during extreme events, they remain consistent with expectations for such scenarios and comparable studies.

Figure 11: Please make missing data white or gray rather than blue, which is also the color of low AODs. It's currently very hard to tell what is missing data vs. low AOD.

Thank you for your suggestion. We have updated Figure 11 to represent missing data in light gray, ensuring a clear distinction from areas of low AOD, which remain blue.

L418: Units of RMSE.

Corrected.

Figure 14 and Tables 2 and 3: Many (most?) readers won't know the EPA regions (at least not precisely). There needs to be a map showing the regions.

Thank you for your suggestion. We have added an EPA region map to Figure 14 to provide clarity and help readers better understand the spatial context of the regions discussed in Tables 2 and 3.

Figure 14: For RF and GWR, are the averages calculated only from the locations of the ground stations (to be consistent with the average of the ground stations)? Or are the averages from RF and GWR calculated from the entire region (in which case, differences with the monitors could be due to differences in the locations used in the averaging)?

The mean surface PM2.5 estimations for RF and GWR in Figure 14 are calculated over the entire region. The differences between our estimations (RF and GWR) and EPA measurements

primarily arise from the spatial distribution of sample points used for averaging. As discussed in the text, RF is more influenced by the spatial distribution of ground stations, with its regional mean values closely following the trends of EPA station measurements. In contrast, GWR is better able to capture PM2.5 variations in regions where stations are more sparsely distributed.

L454-455: Please explain the "FRP < 10000 MW" threshold. MW is an energy rate. How is a rate summed over a year (should the integral of a rate over time be in Joules, not Watts)? Or is it an average rate over the season?

Thank you for pointing this out. You are correct that FRP represents an energy rate, and when summed or integrated over time, it should be expressed in energy units. In this context, the threshold should be described fire radiative energy (unit: MJ), representing the time-integrated fire radiative energy rather than the instantaneous power. We have updated the text to clarify this and corrected the unit to MJ accordingly.

L458-459: I'm not sure how a 17-day investigation alone can give insight into what is happening in the long-term multi-year trend. There is no investigation of how Canadian smoke has changed during the multi-year time period.

To avoid misunderstanding, we revised this sentence to:
Compared with Table [A1], the 17-day investigation highlights how long-range transported smoke from Canada temporarily offsets the descending trend in surface PM2.5 during the study period. For regions 8 to 10, wildfires (including contributions from both local and remote fires) increase the August mean surface PM2.5 by 0 – 97%. While this study focuses on a short-term event, it demonstrates the significant seasonal impact of Canadian smoke on air quality, emphasizing the need for multi-year investigations to assess long-term trends in Canadian smoke contributions.

L467-481: A very important thing missing from these limitations is uncertainties in the plume heights. (1) It affects horizontal transport speed and direction in WRF-Chem, causing errors in the gap filling. (2) It affects the PM:AOD ratio, and if there's variability in plume height between monitors, this will cause errors in the inferred surface concentrations. (3) It will certainly have issues within equation 9 in getting the smoke fractional contribution to PM, as discussed earlier.

Thank you for pointing this out. We have added the uncertainties related to the injection height to section 5.9:
"Uncertainties in the injection height of smoke plumes in WRF-Chem can impact simulation accuracy. Biases in injection height influence horizontal transport speed, direction, and pollution residence time, potentially introducing errors in the AOD gap-filling process. Moreover, uncertainties in the vertical distribution of aerosols can affect the AOD-PM2.5 relationship, with variations in plume height between monitors leading to inaccuracies in estimated surface PM2.5 concentrations."

L476: Figure 4 seems like more monitors than I would expect from FRM alone. Are you sure you aren't using FRM+FEM monitors?

Thank you for pointing this out. Upon reviewing our data, we confirmed that the monitors used correspond to code 88101, which includes both Federal Reference Methods (FRMs) and Federal Equivalent Methods (FEMs). We also clarified this in section 2.2 of the manuscript as follows:

"We obtained the daily surface PM2.5 concentration product that uses Federal Reference Methods and Federal Equivalent Methods (FRMs and FEMs, with code 88101) from US Environmental Protection Agency (EPA) for CONUS within the study period."

L506: particulate *matter*

Corrected.

L522: I don't see how a 17-day study confirms fires becoming the dominant source.
We have revised this to avoid overstating:
Our study highlights the significant contribution of wildfires to particulate pollution during the study period, aligning with prior research that suggests wildfires are becoming an increasingly important source of particulate pollution as industrial pollution declines due to stringent regulations (Xue et al., 2021). However, further multi-year investigations are needed to robustly confirm this trend on a broader temporal scale.

Reviewer 2

This study combined satellite observations, model simulations, and ground measurements to assess the impact of long-term transport smoke from Canadian wildfires on U.S. PM2.5 levels during a 17-day study period. This topic is attractive but the manuscript is not well structed and the presentation of results/conclusion is confusing. The text is hard to read and the figures (too many figures) are in low quality. Detailed comments are listed below

1. The structure of this manuscript needs to be carefully re-design. For example, the first paragraph of introduction talked about PM2.5, fire-related PM2.5, fire-related PM2.5 health effects and again PM2.5 health effects without a logical sequence.

Thank you for pointing this out. We have restructured the first paragraph of the introduction to establish a more logical flow. Additionally, we have reviewed and refined other sections of the manuscript to ensure a more coherent structure throughout.

2. The citation format needs to be double check throughout the text.

Thank you for pointing this out. We have thoroughly reviewed the manuscript to ensure that all citations adhere to the required format and are consistent throughout the text.

3. When averaging 1-km resolution AOD to 0.1-degree grid, how to deal with missing AOD data?

When averaging 1-km resolution AOD to a 0.1-degree grid, we calculate the average using only the valid AOD values within each grid box, excluding any missing values. If a grid box contains no valid AOD values, it is assigned a value of NaN.

4. Description of the AOD gap-filling method is not clear. In the third step, "calculating the ratio of pixels with valid MAIAC AOD over the total pixels", are the total pixels satisfied the pixel selection conditions defined above? If the ratio is less than or equal to 80%, the AOD is filled by geo-weighted method including all the pixels within the pre-selected box or including only the pixels satisfied the pixel selection conditions defined above?

The total pixels in this context refer to all pixels within the selected box, regardless of whether they satisfy the filtering conditions. If the ratio of valid MAIAC AOD pixels to the total pixels in the box is less than 80%, the AOD value is estimated specifically for the target pixel, not for the entire box. Each pixel with missing AOD is evaluated individually using a moving box approach to ensure localized and precise gap-filling. We revised the text:

"To estimate the AOD value for a location without valid MAIAC AOD, we first select a 9×9-pixel box (spatial resolution of 0.1-degree) centered on the target pixel. Within this box, we identify pixels that satisfy two conditions: (a) having valid MAIAC AOD data, and (b) having a small $AOD_{wrf}$ difference (<0.1) compared to the target pixel. If fewer than 50 pixels meet these criteria, the box radius is expanded, and the filtering process is repeated. Once at least 50 pixels are identified, we calculate the ratio of valid MAIAC AOD pixels to the total number of pixels in the box, where total pixels refer to all pixels in the selected box, regardless of filtering. If the ratio exceeds 80%, the AOD for the target pixel is estimated using the ordinary kriging (OK) method, based on the filtered pixels. However, if the ratio is 80% or lower, the AOD is calculated using a geographically weighted regression method that considers the neighboring ratio between MAIAC AOD and $AOD_{wrf}$ . This process is performed individually for each target pixel using a moving box approach."

5. Regarding the hyperparameter of RF model, only the number of tree and number of parameters are considered and the model is over-fitted. The hyperparameter number of leaves (nodes) needs to be optimized to deal with the over fitting issue.

We have optimized the hyperparameters of the RF model, including the number of leaves (nodes), maximum depth of each tree, minimum samples required to split an internal node, and minimum samples required to be in a leaf node, to address the overfitting issue. Through this process, we observed that the GWR method outperformed RF under extreme conditions (e.g., during wildfire events) in our case. The discussion and related performance metrics have been updated accordingly.

6. In section 4.3, the control PM2.5 is calculated from simulated AOD, filled AOD and predicted PM2.5. I am not sure what is the contribution of AOD here. The control PM2.5 can be directly assessed from the estimated PM2.5 by calculating the Canadian fire contribution ratio as the simulated control PM2.5 divided by the simulated true PM2.5. Due to the non-linear relationship between AOD and PM2.5, the AOD-based calculation may introduce larger uncertainty. What's advantage of including AOD in this calculation?

Thank you for raising this point. We agree that the $\frac{PM_{2.5}}{AOD}$ ratio may introduce uncertainty in estimating the smoke contribution to surface PM2.5. To address this, we have revised our method to avoid directly using the $\frac{PM_{2.5}}{AOD}$ ratio. Instead, we now estimate the surface PM2.5 for the control case using the ratio between the WRF-Chem-simulated surface-level P2.5 concentrations of the experiment run (with Canadian smoke) and control run (without Canadian smoke), as follows:

$$PM_{2.5,control} = PM_{2.5} * \left( \frac{PM_{2.5,WRF,control}}{PM_{2.5,WRF,experiment}} \right)$$

The revised method uses the modeled surface P2.5 concentrations from WRF-Chem for both the control and experiment cases, which inherently accounts for the vertical distribution of smoke. By doing so, it avoids the assumption of directly using AOD to estimate surface-level P2.5. We have updated the Section and equation 9, the results in the abstract have also been updated accordingly.

7. The result reported in line 295 directly compared AOD maps in Figure 2 and PM2.5 maps in Figure 4 and discussed the contribution of Canadian wildfires vs. US local fires to surface pollution. However, AOD and PM2.5 are not linearly correlated. This discussion here is with large uncertainty.

Thank you for pointing out the potential uncertainties in directly comparing AOD and PM2.5. To address this concern and enhance clarity, we have included surface PM2.5 dry mass difference maps for the four days in the revised manuscript. These maps directly represent the surface PM2.5 changes attributable to Canadian wildfire emissions, calculated from the difference between the Canadian wildfire case and the control case in the WRF-Chem simulations. By using these difference maps, we can more effectively illustrate the linkage between surface pollution

changes and the associated pressure systems, reducing the uncertainties associated with comparing columnar AOD and surface PM2.5.

8. Section 5.3 discusses the PBL changes at hourly scale with diurnal patterns that is not consistent with other results at daily scale. This section can be removed.

Thank you for the suggestion. We agree that the discussion in Section 5.3 regarding PBL changes at the hourly scale is not fully consistent with the daily-scale analyses presented elsewhere in the manuscript. To maintain coherence and focus, we have removed this section from the revised manuscript.

9. Line 365-370, figure 10 appear early than figure 9 in the manuscript.

We have corrected the order of the figures in the manuscript so that Figure 9 now appears before Figure 10.

10. The different temporal trends reported by local stations, GWR and RF indicating that the model predictions are nor solid. What is the performance of the temporal trends of PM2.5 from EPA stations and from RF/GWR estimations at the same location? Please adjust the model prediction before further analyses.

The mean surface PM2.5 estimations for RF and GWR in Figure 14 are calculated across the entire study region, which inherently includes differences in the spatial distribution of sample points used for averaging. These differences largely explain the discrepancies between the estimations (RF and GWR) and EPA measurements.

As discussed in the text, the RF model tends to be more influenced by the spatial distribution of ground stations, resulting in regional mean values that closely follow the temporal trends of EPA station measurements. In contrast, the GWR model is better suited to capturing PM2.5 variations in areas with sparse station coverage, leading to differences in regional trends compared to RF and EPA measurements.

We believe these differences reflect the strengths and limitations of each model in different spatial contexts rather than an inherent lack of robustness in the predictions.

11. In Table 3, region 1 on Aug-22 with >12.2% Canadian wildfires' PM2.5 effects but only 3.3 $\mu gm^{-3}$5 concentrations? Please double check the number in table 2 and 3.

As mentioned above, we recalculated the surface PM2.5 for the control case. As a result, the percentage and PM2.5 concentration values have been updated in the revised manuscript.

12. Figure 1 What is "r"? Please label it in the figure or in the caption.

Thank you for pointing this out. The "r" in Figure 1 refers to the radius of the selected box used in the analysis. We have updated the figure caption to clarify this.

13. Figure 3 There are too many numbers in the figure.

Thank you for your feedback. To address this, we have changed the surface weather maps in Figure 3 for easier visualization and improved clarity. We hope the updated figure better conveys the key information.

14. Figure 4 Are these purple dots showing missing measurements or showing measurements of 0 µg/m³?

The purple dots in Figure 4 represent measurements less than 2 µg/m³. We have updated the plot to exclude all measurements below this threshold for clarity.

15. Figure 6 What date is this figure about?

Thank you for the comment. Since Section 5.3 has been removed, Figure 6 has also been removed from the revised manuscript.

16. Figure 11 (a), missing AOD should be shown as missing not as 0 in a continuous color scheme.

Thank you for your suggestion. We have updated Figure 11 to represent missing data in light gray, ensuring a clear distinction from areas of low AOD, which remain blue.

17. Figure 12 and Figure 13, the caption should be "100-fold CV" rather than "k-fold CV". Some key parameters, e.g. R and RMSE, should be added in the figures.
Corrected.

18. Figure 14 Why are some data points not on the line?

Thank you for pointing this out. To address your concern, we have remade Figure 14 with improved visualization, including adjustments to the data points and the addition of an EPA region map to provide spatial context.

Reviewer 3

In this paper, the authors used a different approach than what we usually use to estimate the impact of Canadian wildfires on PM2.5 concentrations in the United States. The traditional approach is WRF-CHEM sensitivity simulation. Two simulations are performed with and without

Canadian wildfire emissions. The simulation differences are used to estimate the contribution of Canadian wildfires to the increase in ground PM2.5 concentrations in the United States. In this approach, observational data (ground stations or satellites) are only used to compare with WRF-CHEM results to demonstrate the reliability of model simulations. Therefore, the observational data have little impact on the final results and conclusions.

The highlight of this paper is that the authors incorporate observational data into the estimation, which includes not only ground PM2.5 observations but also satellite AOD observations. The authors first fill the AOD observation gaps caused by satellite limitations through Kriging interpolation and CTM interpolation, and then use two machine learning methods, GWR and RF, combined with PM2.5 ground observations to estimate the daily PM2.5 concentrations at each point in the model simulation area.

I don't have too many questions about the treatment of AOD, but based on Figures 12 and 13, I doubt whether the PM25 surface concentration can be correctly inverted from AOD even with machine learning methods.

As can be seen in Figures 12 and 13, the probability that the 2D PM25 ground concentrations predicted using the 3D columnar data (AOD) match the observed values is low. For example, if the PM25 observations are in the range 0-10, the PM25 predicted by GWR and RF are in the range 0-50 (k-fold plot); the colors representing higher probabilities are overwritten by the colors representing lower probabilities. This may be due to the large markers used by the authors when plotting the graphs. But overall, Figures 12 and 13 are not sufficient to prove the most critical assumptions used in this paper and should be revisited.

Another big problem related to the above discussion is Equation 9. In this equation, the authors claim that the AOD/PM25 relationship derived from WRF-CHEM simulations that include Canadian wildfire emissions can be used for WRF-CHEM simulations that do not include Canadian wildfire emissions. Ground station PM25 observations must be used to derive the AOD/PM25 relationship, but the ground stations only match the model simulation scenario with Canadian wildfire emissions. Since the control runs have no matching observations, the AOD/PM25 relationship in the control is theoretically unsolvable, and using other relationships instead is incorrect.

Apart from these issues, the rest of the paper is well organized and logically clear.

We thank the reviewer for your suggestions.
To address your concerns, we have revised Figures 12 and 13 by reducing the marker sizes, which now clearly display the higher probabilities near the 1:1 line. Most of the predicted PM2.5 values closely align with observations, especially for low-pollution scenarios, demonstrating the model's reliability in such cases. We acknowledge the challenges inherent in using 10 km resolution AOD to predict PM2.5 concentrations at specific points (EPA stations) within a grid. This spatial mismatch can lead to slight overestimation in low-pollution areas. High-pollution events, such as wildfire smoke with large spatial coverage, present additional challenges due to the variability in aerosol properties, vertical distributions, and the dynamic nature of smoke plumes.

To provide additional context for the model's performance, we have included validation metrics such as $R^2$ and RMSE in the figures. These metrics illustrate the model's ability to capture the spatial and temporal patterns of PM2.5 under varying conditions. Our approach aligns with prior studies demonstrating the utility of satellite-derived AOD in estimating surface PM2.5 (Childs et al., 2022; Zhang & Kondragunta, 2021). For instance, during wildfire events, our R values are slightly higher than those reported by (Zhang & Kondragunta, 2021), highlighting the robustness of our method in challenging scenarios.

However, we also recognize the limitations of this approach, particularly for extreme conditions such as wildfire events or for capturing fine temporal resolution dynamics. Future work will focus on refining the methodology to address these limitations and testing the model across diverse scenarios to enhance its robustness and applicability.

For the second problem, we agree that the $\frac{PM_{2.5}}{AOD}$ ratio may not accurately represent the smoke contribution to surface PM2.5 when a significant fraction of smoke resides above the PBL. To address this, we have revised our method to avoid directly using the $\frac{PM_{2.5}}{AOD}$ ratio. Instead, we now estimate the surface PM2.5 for the control case using the ratio between the WRF-Chem-simulated surface-level P2.5 concentrations of the experiment run (with Canadian smoke) and control run (without Canadian smoke), as follows:

$$PM_{2.5,control} = PM_{2.5} * \left( \frac{PM_{2.5,WRF,control}}{PM_{2.5,WRF,experiment}} \right)$$

The revised method uses the modeled surface P2.5 concentrations from WRF-Chem for both the control and experiment cases, which inherently accounts for the vertical distribution of smoke. By doing so, it avoids the assumption of directly using AOD to estimate surface-level P2.5. We have updated the Section and equation 9, the results in the abstract have also been updated accordingly.

Ahern, A., Robinson, E., Tkacik, D., Saleh, R., Hatch, L., Barsanti, K., et al. (2019). Production of secondary organic aerosol during aging of biomass burning smoke from fresh fuels and its relationship to VOC precursors. *Journal of Geophysical Research: Atmospheres, 124*(6), 3583-3606.

Aloyan, A., Arutyunyan, V., Lushnikov, A., & Zagaynov, V. (1997). Transport of coagulating aerosol in the atmosphere. *Journal of aerosol science, 28*(1), 67-85.

Barnard, J. C., Fast, J. D., Paredes-Miranda, G., Arnott, W., & Laskin, A. (2010). Evaluation of the WRF-Chem" Aerosol Chemical to Aerosol Optical Properties" Module using data from the MILAGRO campaign. *Atmospheric Chemistry and Physics, 10*(15), 7325-7340.

Binkowski, F. S., & Shankar, U. (1995). The regional particulate matter model: 1. Model description and preliminary results. *Journal of Geophysical Research: Atmospheres, 100*(D12), 26191-26209.

Carrico, C. M., Kreidenweis, S. M., Malm, W. C., Day, D. E., Lee, T., Carrillo, J., McMeeking, G. R., & Collett Jr, J. L. (2005). Hygroscopic growth behavior of a carbon-dominated aerosol in Yosemite National Park. *Atmospheric environment, 39*(8), 1393-1404.

Childs, M. L., Li, J., Wen, J., Heft-Neal, S., Driscoll, A., Wang, S., et al. (2022). Daily local-level estimates of ambient wildfire smoke PM2. 5 for the contiguous US. *Environmental Science & Technology, 56*(19), 13607-13621.

Crippa, P., Sullivan, R. C., Thota, A., & Pryor, S. C. (2017). The impact of resolution on meteorological, chemical and aerosol properties in regional simulations with WRF-Chem. *Atmospheric Chemistry and Physics, 17*(2), 1511-1528.

Emerson, E. W., Hodshire, A. L., DeBolt, H. M., Bilsback, K. R., Pierce, J. R., McMeeking, G. R., & Farmer, D. K. (2020). Revisiting particle dry deposition and its role in radiative effect estimates. *Proceedings of the National Academy of Sciences, 117*(42), 26076-26082.

Fu, Y., Li, R., Wang, X., Bergeron, Y., Valeria, O., Chavardès, R. (2020). Fire detection and fire radiative power in forests and low-biomass lands in Northeast Asia: MODIS versus VIIRS Fire Products. *Remote Sensing, 12*(18), 2870.

Gomez, S. L., Carrico, C., Allen, C., Lam, J., Dabli, S., Sullivan, A., et al. (2018). Southwestern US biomass burning smoke hygroscopicity: The role of plant phenology, chemical composition, and combustion properties. *Journal of Geophysical Research: Atmospheres, 123*(10), 5416-5432.

Junghenn Noyes, K. T., Kahn, R. A., Limbacher, J. A., & Li, Z. (2021). Canadian and Alaskan wildfire smoke particle properties, their evolution, and controlling factors, from satellite observations. *Atmospheric Chemistry and Physics Discussions, 2021*, 1-34.

Powers, J. G., Klemp, J. B., Skamarock, W. C., Davis, C. A., Dudhia, J., Gill, D. O., et al. (2017). The weather research and forecasting model: Overview, system efforts, and future directions. *Bulletin of the American Meteorological Society, 98*(8), 1717-1737.

Ryu, Y. H., & Min, S. K. (2022). Improving Wet and Dry Deposition of Aerosols in WRF-Chem: Updates to Below-Cloud Scavenging and Coarse-Particle Dry Deposition. *Journal of Advances in Modeling Earth Systems, 14*(4), e2021MS002792.

Sun, H., Jiao, R., Xu, H., An, G., & Wang, D. (2019). The influence of particle size and concentration combined with pH on coagulation mechanisms. *Journal of Environmental Sciences, 82*, 39-46.

Val Martin, M., Kahn, R. A., & Tosca, M. G. (2018). A global analysis of wildfire smoke injection heights derived from space-based multi-angle imaging. *Remote Sensing, 10*(10), 1609.

Xue, Z., Gupta, P., & Christopher, S. (2021). Satellite-based estimation of the impacts of summertime wildfires on PM 2.5 concentration in the United States. *Atmospheric Chemistry and Physics, 21*(14), 11243-11256.

Ye, X., Arab, P., Ahmadov, R., James, E., Grell, G. A., Pierce, B., et al. (2021). Evaluation and intercomparison of wildfire smoke forecasts from multiple modeling systems for the 2019 Williams Flats fire. *Atmospheric Chemistry and Physics Discussions, 2021*, 1-69.

Zhang, H., & Kondragunta, S. (2021). Daily and hourly surface PM2. 5 estimation from satellite AOD. *Earth and Space Science, 8*(3), e2020EA001599.

---

## Author Response (AR2)

Could the authors make the same graphs as Figures 10 and 11 using PM25 from the WRF experimental run as a reference? Because the basic assumption of this paper is that the derived PM25 data is better than the model simulation data.

Thank you for the suggestion. We have added a new figure comparing WRF-Chem simulated PM2.5 with EPA ground measurements. This comparison, presented after the GWR method results, highlights the limitations of standalone chemical transport models and further reinforces the value of applying GWR.